# Kinesin-3 mediated axonal delivery of presynaptic neurexin stabilizes dendritic spines and postsynaptic components

Devyn Oliver[1], Shankar Ramachandran[1], Alison Philbrook[1], Christopher M. Lambert[1], Ken C. Q. Nguyen[2], David H. Hall[2], Michael M. Francis[1]*

1 Department of Neurobiology, University of Massachusetts Chan Medical School, Worcester, Massachusetts, United States of America, 2 Department of Neuroscience, Albert Einstein College of Medicine, New York, New York, United States of America

* michael.francis@umassmed.edu

**Data Availability Statement:** All relevant data are within the manuscript and its Supporting Information files.

## Abstract

The functional properties of neural circuits are defined by the patterns of synaptic connections between their partnering neurons, but the mechanisms that stabilize circuit connectivity are poorly understood. We systemically examined this question at synapses onto newly characterized dendritic spines of *C. elegans* GABAergic motor neurons. We show that the presynaptic adhesion protein neurexin/NRX-1 is required for stabilization of postsynaptic structure. We find that early postsynaptic developmental events proceed without a strict requirement for synaptic activity and are not disrupted by deletion of neurexin/*nrx-1*. However, in the absence of presynaptic NRX-1, dendritic spines and receptor clusters become destabilized and collapse prior to adulthood. We demonstrate that NRX-1 delivery to presynaptic terminals is dependent on kinesin-3/UNC-104 and show that ongoing UNC-104 function is required for postsynaptic maintenance in mature animals. By defining the dynamics and temporal order of synapse formation and maintenance events *in vivo*, we describe a mechanism for stabilizing mature circuit connectivity through neurexin-based adhesion.

## Author summary

The nervous system is composed of networks of interconnecting cells called neurons. Within these networks, individual neurons establish connections or synapses with partnering neurons. Key connections within these networks must be stabilized and preserved throughout the lifetime of an animal in order for the nervous system to function properly. Alterations in synapse connectivity and stability are linked with numerous neurodevelopmental and neurodegenerative diseases. Therefore, it is crucial to gain a complete understanding of the steps in assembly of a developing synapse, and of the mechanisms involved in synapse maturation and maintenance. Using microscopy and live imaging approaches at newly characterized spiny synapses in a genetic model organism, the nematode *Caenorhabdtis elegans*, we describe the sequence of events by which developing synapses are constructed. In addition, we show that the cell adhesion molecule, neurexin, is

**Funding:** This research was supported by NIH
NINDS RO1NS064263 (MMF), R21NS101649
(MMF), NIH OD 01943 (DHH), and F31NS103365
(DO). The Philips CM10 TEM electron microscope
used in this study was acquired through a NIH
Shared Instrumentation Grant (1S10OD016214-
01A1) at Albert Einstein College of Medicine.

**Competing interests:** The authors have declared
that no competing interests exist.

required for stabilization of these synapses into adulthood. In the absence of neurexin,
neuronal connections begin to form but then disappear. A specific motor protein, called
kinesin-3, is required to deliver neurexin to synapses in order to stabilize specific connec-
tions between neurons. An improved understanding of the sequence of events in synapse
formation and how neurexin is delivered to developing synapses for stabilizing connec-
tions may help to advance our understanding of disorders and diseases that arise from dis-
ruptions of these processes.

## Introduction

The capabilities of neural circuits to perform specific functions arise from the patterns of syn-
aptic connections between their partnering neurons. The organization of these connections is
circuit-specific and established through a complex process that involves the coordinated
assembly and maturation of specialized pre- and postsynaptic structures on appropriate part-
nering neurons, and their maintenance in the mature nervous system. We now have a general-
ized understanding of synapse structure. Active zone (AZ) proteins and neurotransmitter-
filled synaptic vesicles position near the presynaptic membrane for rapid release while neuro-
transmitter receptors are clustered at high density in apposition to these sites in order to
ensure the fidelity of synaptic communication. Genetic studies have identifed numerous muta-
tions that alter circuit connectivity or affect the overall structural organization of synapses [1–
3]. However, for many of the synapse-associated proteins affected by these mutations, we do
not yet have a mechanistic understanding of their roles in establishing synaptic connections,
or how their disruption may lead to alterations in synapse stability. Gaining an enhanced
understanding of the sequence of events involved in synapse assembly, maturation, and main-
tenance, and their relative timing *in vivo* is critical for addressing these questions.

Several prior studies have examined molecular events during synaptogenesis. *In vivo* studies
in *C. elegans* and *Drosophila* have largely focused on the formation of the presynaptic active
zone. Collectively, these studies provide compelling evidence for a model where active zone
assembly occurs sequentially. While there is some variability across synapse type, the early
stages of this process are generally organized by the highly conserved synaptic scaffolds SYD-
2/Liprin-α and SYD-1/Rho GTPase, which then recruit additional key conserved AZ proteins
such as ELKS-1/Bruchpilot, Piccolo family members, and UNC-10/RIM for subsequent stages
of assembly, including clustering of $Ca^{2+}$ channels, and the recruitment of synaptic vesicles
[2,4–7]. Remarkably, nascent AZs can assemble quite rapidly (within minutes) but then often
undergo a more extended period of maturation that can last for several hours [5,8–10]. The
relationship of postsynaptic development to presynaptic assembly remains less clear. Live
imaging studies of cultured mouse hippocampal neurons have suggested that the assembly of
presynaptic components precedes recruitment of postsynaptic receptors and scaffolds [11].
More recently, 2-photon imaging of organotypic slice cultures from rat hippocampus showed
that new spines accumulate glutamate receptors concurrently with their growth and are com-
petent to participate in transmission within a few hours after outgrowth [12]. Comparatively
few *in vivo* studies have investigated coordinated pre- and postsynaptic development, and sig-
nificant questions remain about molecular mechanisms that direct linkages between pre- and
postsynaptic specializations to promote their maturation and stabilization.

Evolutionarily conserved synaptic adhesion proteins, such as neurexins, are prime candi-
dates for coordinating pre- and postsynaptic events during synaptogenesis. Their importance
is underscored by the fact that neurexin alterations are associated with cognitive disease,

including schizophrenia and autism spectrum disorders [13,14]. Neurexins are typically localized presynaptically and linked to postsynaptic binding partners such as dystroglycans, LRRTMs, neuroligins, and cerebellins, through extracellular laminin and EGF-like repeats to establish transsynaptic connections. Neurexin mediated transsynaptic signaling has been implicated in key aspects of synapse development and function. For example, loss of presynaptic neurexin at the fly neuromuscular junction increases the length of the presynaptic density, and also alters the size and molecular composition of apposed muscle glutamate receptor clusters [15,16]. The broad brain distribution of neurexins and the many neurexin isoform variants in the nervous system enable complex, cell-type specific functions for neurexins at synapses [17,18]. For example, conditional deletion of mouse neurexins revealed strikingly divergent functions across synapses formed by cortical inhibitory interneurons and those formed by cerebellar climbing fibers onto Purkinje neurons [19]. Importantly, molecular mechanisms for the trafficking and delivery of neurexins to synapses remain incompletely defined. Genetic tools that enable cell-type specific analysis are therefore critical for uncovering precise functional roles of synaptic adhesion proteins and mechanisms for their regulation within the context of individual neural circuits.

We previously identified finger-like protrusions from the dendritic processes of *C. elegans* DD GABAergic motor neurons (**Fig 1A**) [20]. Characterization of these structures by our laboratory and others pointed towards the idea that they receive synaptic input from presynaptic cholinergic motor neurons and serve analogous roles to dendritic spines in the mammalian brain [21–23]. Deletion of *nrx-1*, the sole *C. elegans* ortholog of neurexin, disrupts these dendritic spines and impairs proper localization of cholinergic receptor clusters to postsynaptic sites on GABAergic dendrites [20]. Here we define the order and timing of pre- versus postsynaptic development at spine-associated synapses and elucidate the role of neurexin in the coordination of these processes *in vivo*. We find that clusters of both presynaptic proteins and postsynaptic receptors are clearly visible prior to spine outgrowth. In the absence of neurexin, immature spines and receptor clusters form initially but then disappear, indicating that neurexin is required for synapse stabilization and maturation rather than synaptogenesis. These maintenance and maturation processes are supported by kinesin-3/UNC-104 motor dependent transport of NRX-1 to presynaptic terminals. Together, our results suggest that axonal delivery of neurexin to sites of synapse formation is critical for stabilization of mature postsynaptic structures.

## Results

We initially explored morphological features of mature spines using 3D rendering of spines from confocal imaging and by electron microscopy. We observed considerable morphological diversity using both approaches, as noted in prior work (**Figs 1A, S1A and S2**) [21,23,24]. To explore how the neuronal cytoskeleton may contribute to spine morphological features, we examined the localization of F-actin and tubulin in GABAergic DD dendrites using DD neuron-specific expression of GFP::UtrCH (GFP fused to the Utrophin calponin homology domain) [25] and GFP::TBA-1 (GFP-tagged α-tubulin) [26] respectively. We found that the two markers largely occupy distinct territories, where F-actin is highly localized to dendritic spines (**Fig 1B and 1C**) while tubulin occupies the dendritic shaft (**Fig 1D and 1E**) with more variable localization near the spine base (**S1B Fig**).

We previously showed stimulation of cholinergic motor neurons elicited Ca$^{2+}$ responses in GABAergic motor neurons [20]. Here we asked whether stimulation of presynaptic cholinergic neurons was sufficient to elicit calcium responses in GABAergic DD dendrites. We performed *in vivo* calcium imaging of evoked responses in spines using combined expression of a

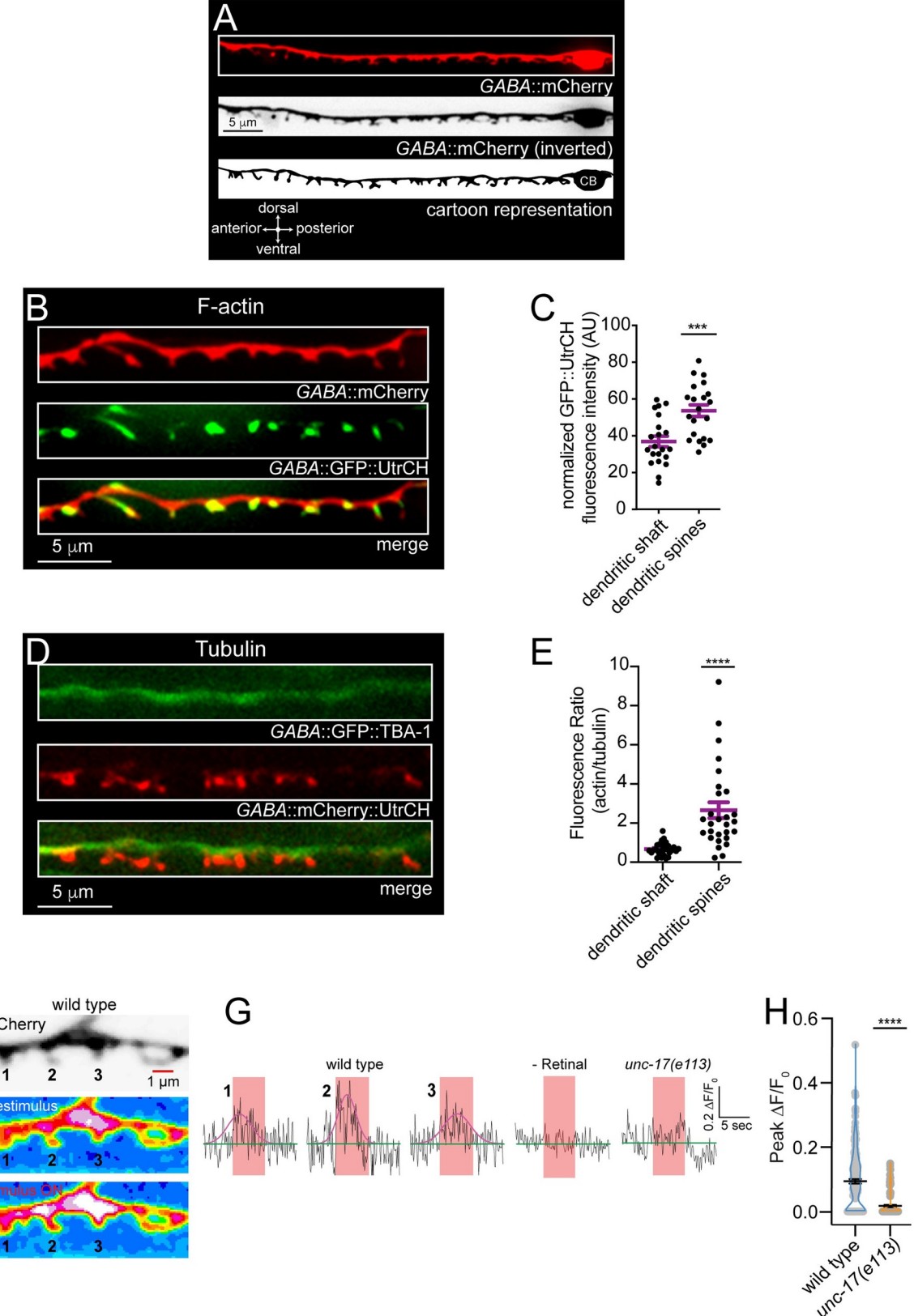

**Fig 1. *C. elegans* GABAergic DD neuron dendritic spines share characteristics with dendritic spines in vertebrates.** (A) Representative fluorescent images and cartoon representation of the ventral nerve cord/dendrite of a single L4 stage GABAergic DD (DD2) motor neuron with ventrally projecting dendritic spines. CB, cell body. (B) Representative images of DD dendritic spines from animal co-expressing P*flp-13*::mCherry (labeling dendrite, red) and P*flp-13*::GFP::UtrCH (labeling F-actin, green). F-actin is highly enriched in DD dendritic spines. (C) Scatterplot of F-actin/GFP::UtrCH fluorescence in dendritic shaft compared to dendritic spines, normalized to total fluorescence within the ROI. Bars indicate mean ± SEM. Student's t-test, ***$p<0.001$, n = 21 animals. (D) Fluorescent images of tubulin (P*flp-13*::GFP::TBA-1) (green) and actin (P*flp-13*::mCherry::UtrCH) (red) in DD neurons. Dendritic spines are highly enriched with F-actin while tubulin occupies the main dendritic process. (E) Quantification of fluorescence ratio of actin/tubulin in the dendritic shaft versus dendritic spines. Student's t-test, ****$p<0.0001$, n = 31 spines from 6 animals. (F) Calcium transients recorded from wild type GABAergic DD motor neuron dendrites expressing myrGCaMP6f (P*flp-13*::myrGCaMP6f::SL2::mCherry) during stimulation of presynaptic cholinergic motor neurons (P*acr-2*::Chrimson). Top, representative image showing dendritic spines identified by mCherry fluorescence (inverted LUT). Representative maximum intensity projection heat maps showing calcium fluxes immediately prior to stimulation (prestimulus, middle panel) and during stimulus (ON, bottom panel). Heat maps were generated by maximum intensity projection of myrGCaMP6f fluorescence from the first 4 seconds of recording (prestimulus) and the 5 s period of stimulation (red shaded bars in G). Numbering corresponds to representative traces in Fig 1G. (G) Representative evoked responses for indicated dendritic spines in wild type animals (wild type 1, 2 and 3, Fig 1F), animals grown in the absence of retinal (-Retinal), and *unc-17(e113)* mutants. Calcium responses were continuously recorded at 10 Hz for 15 s. Prestimulus calcium fluxes were recorded for 5 s, followed by stimulation (625 nm, ~30 mW/cm$^2$) of cholinergic motor neurons for 5 s. All data was normalized to prestimulus ($\Delta F/F_0$). Red shaded bar indicates duration of stimulation. Pink traces indicate Gaussian fits to responses ($\Delta F/F_0$) during stimulation. (H) Scatter plot showing peak $\Delta F/F_0$ of responses during stimulation in wild type or *unc-17(e113)* mutants. Bars, mean ± SEM. Statistical analysis, Student's t-test, ****$p<0.0001$. n $\geq$ 10 animals.

membrane-associated GCaMP6f calcium sensor in DD GABAergic neurons and a red shifted channelrhodopsin, Chrimson, in cholinergic neurons [20]. Following 5 seconds of baseline recording (488 nm, 100 ms exposure), we measured calcium responses to presynaptic depolarization (5 s, 625 nm, 30 mW/cm$^2$) (**Figs 1F–1H and S3**). We noted significant fluorescence increases during the stimulation window. These occurred solely in the presence of retinal and typically appeared simultaneously across multiple spines, returning to baseline within 5 s following stimulation. Importantly, mutation of the cholinergic vesicular transporter/VAChT *unc-17*, reduced evoked calcium responses by 81%, indicating the Ca$^{2+}$ responses we observed in dendrites were largely dependent on presynaptic acetylcholine release (**Fig 1G and 1H**). Our findings are consistent with those of another recent study [23] indicating that spines on DD GABAergic neurons are major sites of synaptic input.

Mitochondria and other organelles are often localized near postsynaptic specializations in order to sustain synaptic function [27,28]. We examined the distribution of organelles within DD dendrites using fluorescent reporters labeling Golgi (AMAN-2::GFP) [29], mitochondria (Pre-Su9::GFP) [30], and endoplasmic reticulum (ER) (RFP::TRAM-1) [29,31]. We found that Golgi is exclusively labeled in the DD cell bodies (**S1C Fig**). In contrast, mitochondria and rough ER markers are present in the main dendritic shaft near spines (**S1D–S1G Fig**), perhaps suggesting involvement in dendritic signaling though we cannot fully exclude potential contributions of overexpression [32,33]. Mitochondria were also evident in dendritic processes near spines by electron microscopy (**S2B–S2D Fig**).

## The development of synapses at DD GABAergic spines

We next investigated the relative timing of developmental events during the formation of synapses onto spines. *C. elegans* progress through four larval stages of development (L1-L4) prior to adulthood. DD neurons undergo a well-characterized program of synaptic remodeling such that the mature circuit organization is established during the transition from L1 to L2 stage [34,35]. The formation of new synaptic connections between postembryonic born cholinergic neurons and the ventral dendrites of DD GABAergic neurons offers a window to investigate *de novo* formation of neuron-neuron synapses. We focused on understanding the time course of spine formation relative to 2 key events in synaptogenesis: 1) the formation of presynaptic release sites, and 2) the clustering of postsynaptic receptors and F-actin.

The appearance of newly born ventral cholinergic motor neurons progressed anteriorly to posteriorly starting roughly 16 hours after hatch, consistent with the birth and integration of cholinergic neurons during this time frame (S4 Fig) [36]. We analyzed the distribution of pre- and postsynaptic markers at timepoints before (12 hours after hatch), during (16, 18, 20 hours after hatch) and after (24, 32 hours after hatch, L4 stage/42-50 hours) DD synaptic remodeling (Figs 2, S4 and S5). We first assessed the formation of the active zone and presynaptic specialization by analyzing clustering of the Piccolo-like active zone protein, CLA-1, (GFP::CLA-1e) [37] and the synaptic vesicle-associated protein synaptobrevin/SNB-1 (SNB-1::GFP) (Figs 2A, 2D–2E and S5). Overall, we found CLA-1 is more discretely localized to putative synapses than SNB-1, both at earlier timepoints and throughout development, consistent with its specific localization to active zones. Immediately prior to remodeling (~12 hours after hatch), we did not observe significant localization of either presynaptic CLA-1 or SNB-1 adjacent to the DD process. Shortly after (16 hours after hatch), we noted the initial appearance of SNB-1- and CLA-1-associated fluorescence in presynaptic cholinergic processes. Over the next 4 hours (20 hours after hatch) individual CLA-1 clusters were more clearly distinguishable, suggesting CLA-1 association with developing active zone structures (Figs 2A, 2D, 2L and S5A). Synaptic vesicle fluorescence was initially diffuse at 16 hours after hatch and became more clearly organized into discrete puncta over a similar time course to CLA-1 (Figs 2A, 2E, 2L and S5B). We did not observe the emergence of DD dendritic spines until 24–32 hours after hatch, well after initial active zone formation and recruitment of synaptic vesicles (Figs 2A, 2C–2E and S5A and S5B). Spines continued to mature through L4 stage, increasing in both length and number (S4D–S4F Fig).

We next examined the clustering of postsynaptic acetylcholine receptors in DD dendrites (ACR-12::GFP). Prior to remodeling (12 hours after hatch), we did not observe detectable levels of cholinergic receptors in ventral DD processes. Surprisingly, we noted immature receptor clusters faintly visible in the dendritic shaft by 16 hours after hatch, well prior to the emergence of dendritic spines (Figs 2B, 2F, 2L and S5C). These receptor clusters increased in number and redistributed towards the tips of growing spines by 32 hours (Figs 2B, 2F, 2L and S5C). We found a similar developmental trend using DD neuron-specific labeling of the LEV-10 transmembrane auxiliary protein [38], previously shown to concentrate in spines [39] (S6A and S6B Fig). By L4 stage, receptor and LEV-10 clusters are clearly visible at the tips of mature spines. Our analysis indicates that the initial stages of development of postsynaptic (AChR and LEV-10 clusters) structures occur prior to spine outgrowth, raising questions about how these initial processes may be regulated.

To begin to address this question, we analyzed the distribution of F-actin in dendrites of the DD neurons during synapse formation (Figs 2B, 2G–2L and S5D). We noted that clusters of F-actin were evident in the ventral DD processes prior to the completion of synaptic remodeling (12 hours after hatch), prior to presynaptic CLA-1 and synaptic vesicle accumulation, and before the clustering of postsynaptic receptors. Dendritic F-actin-based structures became more abundant coincident with increases in the number of presynaptic CLA-1 and postsynaptic receptor clusters. To investigate this process in real-time, we used live imaging to examine the dynamics of postsynaptic F-actin (GFP::UtrCH) in the developing DD dendrite. We found that F-actin is highly dynamic during early developmental stages (16–20 hours after hatch) compared to L4 stage where the circuit has completed maturation (S6C–S6H Fig and S1 and S2 Videos). In young animals, GFP::UtrCH clusters often shuttled out of the cell body to the main dendritic process, perhaps indicating redistribution of F-actin to sites of postsynaptic assembly. By 24 hours after hatch, we observed clear co-localization of F-actin with newly formed AChR clusters in the dendritic shaft. By L4 stage, AChR clusters are stably sequestered at the tips of spines, while F-actin occupies the spine head and neck (Fig 2H–2K).

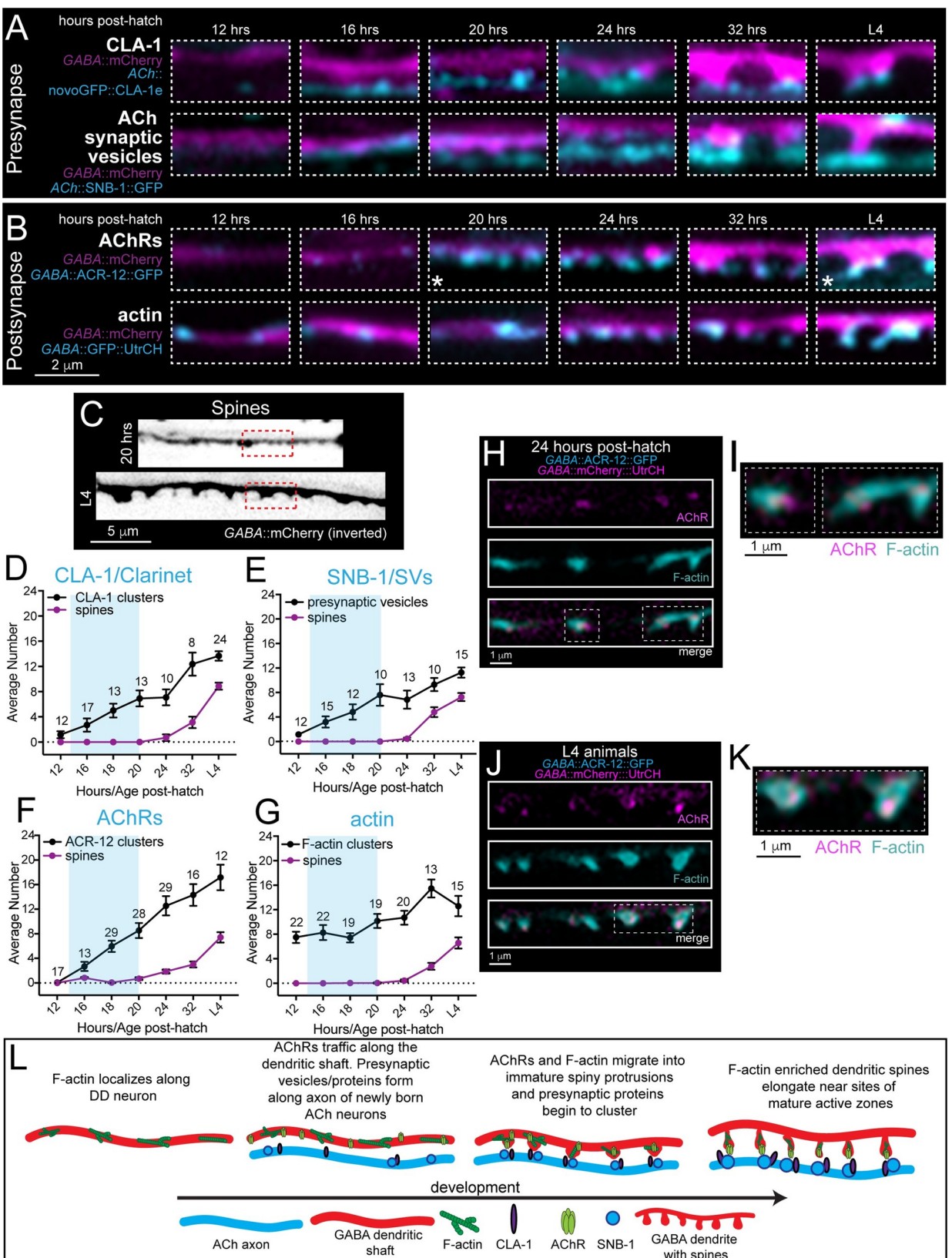

**Fig 2. Temporal order of molecular events during synapse and spine development.** (A) Fluorescent images of GABAergic dendrites at 12, 16, 20, 24, and 32 hours after hatch and at L4 stage (~42–50 hours post-hatch). Animals express a GABAergic dendrite marker (P*flp-13*::mCherry) with either presynaptic active zone (P*unc-17β*::GFPnovo2::CLA-1e) (top) or synaptic vesicle marker (P*acr-5*::SNB-1::GFP) (bottom). Images are pseudo colored to indicate spines (magenta) and presynaptic CLA-1 or vesicle clusters (cyan). (B) Fluorescent images of GABAergic dendrites at 12, 16, 20, 24, and 32 hours after hatch and L4 animals (~42–50 hours post-hatch). Animals express a GABAergic dendrite marker (P*flp-13*::mCherry) with either postsynaptic AChR (P*unc-47*::ACR-12::GFP, top) or postsynaptic actin (P*flp-13*::GFP::UtrCH, bottom) markers. Images are pseudo colored to indicate spines (magenta) and postsynaptic AChR or F-actin (cyan). White asterisks (*) indicate images used in Fig 2C. (C) Fluorescent images (inverted LUT) of GABAergic dendrites at 20 hours after hatch and L4 stage (~42–50 hours post-hatch) as indicated. Animals express a GABAergic dendrite marker (P*flp-13*::mCherry). Dashed red boxes indicate regions used in Fig 2B (asterisks). (D-G) Quantification of spine number (purple) relative to clusters of active zone protein/CLA-1 (D), synaptic vesicle/SNB-1 (E), AChR/ACR-12 (F), and F-actin/UtrCH (G) (black) through development. Clusters of each of these markers are visible well prior to spine outgrowth. Blue shading indicates the approximate period of DD remodeling. For this and subsequent figures, numbering indicates animals quantified at each timepoint for each genotype. Data points and bars indicate mean ± SEM. (H) Fluorescent confocal images of DD1 AChR/ACR-12 (P*flp-13*::ACR-12::GFP, magenta) and F-actin clusters (P*flp-13*::mCherry::UtrCH, cyan) 24 hours after hatch. Dotted rectangle indicates high magnification image shown in 2I. (I) Expanded view of DD1 dendrites 24 hours after hatch from Fig 2H. (J) Fluorescent confocal images of DD1 AChR/ACR-12 (P*flp-13*::ACR-12::GFP, magenta) and F-actin clusters (P*flp-13*::mCherry::UtrCH, cyan) at L4 stage. Dotted rectangle indicates high magnification image shown in 2K. (K) Expanded view of DD1 dendrites 24 hours after hatch from Fig 2J. (L) Cartoon summary of the order of events during development of synapses onto DD neurons. In early development coding, presynaptic vesicles (SNB-1) and active zone protein (CLA-1) localize along presynaptic terminals while postsynaptic F-actin and postsynaptic receptors cluster along the postsynaptic process. Following their formation, dendritic spines begin to extend. Spines continue to elongate throughout development and into the mature circuit (L4).

## Dendritic spine formation does not strongly require synaptic activity

Our above analysis indicated that the localization of synaptic vesicles and active zone proteins, such as CLA-1, occur prior to spine outgrowth. In the rodent brain, spine morphogenesis is clearly regulated by presynaptic activity [40–43], but recent evidence suggests initial spine outgrowth may proceed independently of synaptic activity [44–46]. We therefore next asked whether presynaptic cholinergic activity is important for the formation of DD dendritic spines. To address this question, we analyzed the number of dendritic spines (P*flp-13*::mCherry or P*flp-13*::myrGFP) and cholinergic receptors (P*flp-13*::ACR-12::GFP) at L4 stage in strains carrying mutations that affect various aspects of synaptic function and neuronal excitability (**Table 1** and **S7 Fig**). Surprisingly, the abundance of spines and receptor clusters at L4 stage were not significantly affected by disruption of ACh synthesis in cholinergic motor neurons (mutation of the vesicular acetylcholine transporter *unc-17*) or by a strong reduction in synaptic vesicle exocytosis (mutation of the syntaxin binding protein *unc-18* or the fusion regulator *unc-13*). Similarly, loss-of-function mutations in genes required for dense core vesicle release (CAPS/*unc-31*) and Ca²⁺ signaling (VGCC/*unc-2*) did not significantly reduce the number of spines or receptor clusters in the mature circuit (**Table 1** and **S7 Fig**). Our data indicate that severe alterations in synaptic activity have very limited effects on the abundance of L4 stage dendritic spines. However, we cannot exclude the possibility of developmental delays in spine formation [23]. Notably, many of the mutations we examined significantly altered spine length and the size of receptor clusters (**Table 1**), suggesting that activity may influence other aspects of spine morphology and postsynaptic structure. Together, our findings point toward a model where spine formation and postsynaptic development proceed without a strong requirement for synaptic activity, while subsequent spine morphogenesis is more clearly affected.

## Presynaptic NRX-1 stabilizes postsynaptic components

In previous work, we found that mature spines are severely reduced in L4 stage *nrx-1(wy778)* mutants [20]. We found a similarly severe reduction in *nrx-1(wy1155)* null mutants where the entire *nrx-1* coding region is deleted [47], indicating that both alleles are nulls and highlighting the potential importance of activity-independent mechanisms for spine formation (**S7D and S7E Fig**). Mutation of *nrx-1* also disrupted dendritic calcium responses evoked by presynaptic stimulation of cholinergic motor neurons (**S8 Fig**). To assess involvement of presynaptic

**Table 1. Analysis of dendritic spines and cholinergic receptors in mutant strains with altered synaptic activity.** Dendritic spine number, spine length, ACR-12/AChR cluster number, and AChR cluster size at L4 stage normalized to the same measurements from wild type are shown (mean ± SEM) for each genotype (ACR-2/nAChR UNC-2/CaV2α, UNC-13/MUNC-13, UNC-17/VAChT, UNC-18/MUNC-18, UNC-31/CAPS). Mutations that affected synaptic activity had little effect on the density of spines or AChR clusters, but had variable effects on spine length and AChR cluster size. One-way AVOVA, Dunnett's multiple comparisons test, *$p<0.05$, **$p<0.01$, ***$p<0.001$, ****$p<0.0001$.

| Genotype | Allele type | Spines | | | | | | AChR/receptors | | | | |
|---|---|---|---|---|---|---|---|---|---|---|---|---|
| | | Normalized spine number ± SEM (%) | N | P value | Normalized spine length SEM (%)± | N | P value | Normalized ACR-12 cluster number ± SEM (%) | N | P value | Normalized ACR-12 cluster size ± SEM (%) | N | P value |
| wild type | | 100.0 ± 3.5 | 73 | ns | 100.0 ± 1.8 | 499 | ns | 100.0 ± 6.3 | 17 | ns | 100.0 ± 5.1 | 220 | ns |
| *acr-2 (ok1887)* | null [75] | 86.2 ± 9.1 | 24 | ns | 77.1 ± 2.4 | 141 | **** | 94.6 ± 10.4 | 20 | ns | 109.8 ± 5.9 | 245 | ns |
| [1]*acr-2 (n2420)gf* | gain of function [75] | 52.9 ± 5.9 | 26 | **** | 83.8 ± 2.6 | 135 | ** | 75.6 ± 6.3 | 18 | ns | 81.3 ± 5.4 | 178 | ns |
| *unc-2 (e55)* | null [76] | 81.8 ± 7.6 | 29 | ns | 90.9 ± 3.7 | 122 | ns | 77.2 ± 9.5 | 18 | ns | 83.9 ± 5.7 | 172 | ns |
| *unc-2 (zf35)gf* | gain of function [77] | 81.7 ± 7.2 | 22 | ns | 114.7 ± 4.8 | 117 | * | 120.0 ± 7.2 | 20 | ns | 130.1 ± 7.2 | 312 | ** |
| [2]*unc-13 (e51)* | hypomorph [78] | 62.1 ± 8.1 | 18 | ** | 121.5 ± 7.4 | 78 | *** | 85.0 ± 10.6 | 19 | ns | 74.8 ± 4.4 | 210 | ns |
| *unc-13 (s69)* | severe hypomorph [3,79] | 96.2 ± 4.1 | 21 | ns | 95.8 ± 3.1 | 128 | ns | | | | | | |
| *unc-17 (e113)* | null (personal communication from J. Rand) | 104.5 ± 6.8 | 12 | ns | 94.8 ± 4.4 | 83 | ns | 101.6 ± 7.1 | 24 | ns | 119.8 ± 5.7 | 317 | ns |
| *unc-17 (e245)* | hypomorph [80] | 73.8 ± 8.4 | 17 | ns | 82.8 ± 3.1 | 71 | ns | | | | | | |
| *unc-18 (e234)* | putative null [81] | 76.9 ± 10.0 | 11 | ns | 98.5 ± 3.9 | 59 | ns | 96.4 ± 9.9 | 19 | ns | 113.3 ± 6.5 | 238 | ns |
| *unc-31 (e169)* | putative null [82] | 97.6 ± 5.7 | 22 | ns | 106.3 ± 3.6 | 166 | ns | 107.0 ± 8.8 | 21 | ns | 123.9 ± 5.9 | 291 | * |

[1]Dendritic spines were unaffected in *acr-2(ok1887)* null animals, but were reduced in *acr-2(gf)* animals. Previously described gene expression changes in cholinergic motor neurons of *acr-2(gf)* animals may account for this observation [83].

[2]Though spines were significantly reduced in *unc-13(e51)* mutants, this effect appeared specific for this allele. We did not observe a significant reduction in *unc-13(s69)* null mutants or in *unc-17(e113)* and *unc-18(e234)* mutants where ACh release is severely reduced.

NRX-1 in the establishment of spines, we asked when NRX-1 first localizes to presynaptic terminals of cholinergic motor neurons. Axonal clusters of NRX-1::GFP are present by 16 hours after hatch (**Fig 3A and 3B**), approximately the same time at which assemblies of the presynaptic scaffold CLA-1 become visible (**Fig 2**). Diffuse synaptic vesicle material is also visible at this time (**Fig 2**) but has yet to be organized into discrete clusters. Notably, presynaptic NRX-1::GFP clusters and spines are similar in number by L4 stage, while presynaptic CLA-1 and SNB-1 clusters appear more abundant than spines (**Fig 2D and 2E**). This may suggest that a subset of presynaptic release sites lacking NRX-1 are not associated with dendritic spines; however some of these presynaptic puncta may represent trafficking vesicles rather than mature release sites, complicating interpretation.

The early arrival of NRX-1 to synaptic terminals could indicate a role in either initial synapse formation or subsequent stabilization and maturation. To distinguish between these possibilities, we analyzed the development of dendritic spines and cholinergic receptor clusters in *nrx-1(wy778)* null mutants. We quantified spine and AChR cluster number over a similar time course as previously completed for wild type. Surprisingly, we noted that immature spines and receptor clusters were evident in *nrx-1* mutants during early development, albeit at slightly

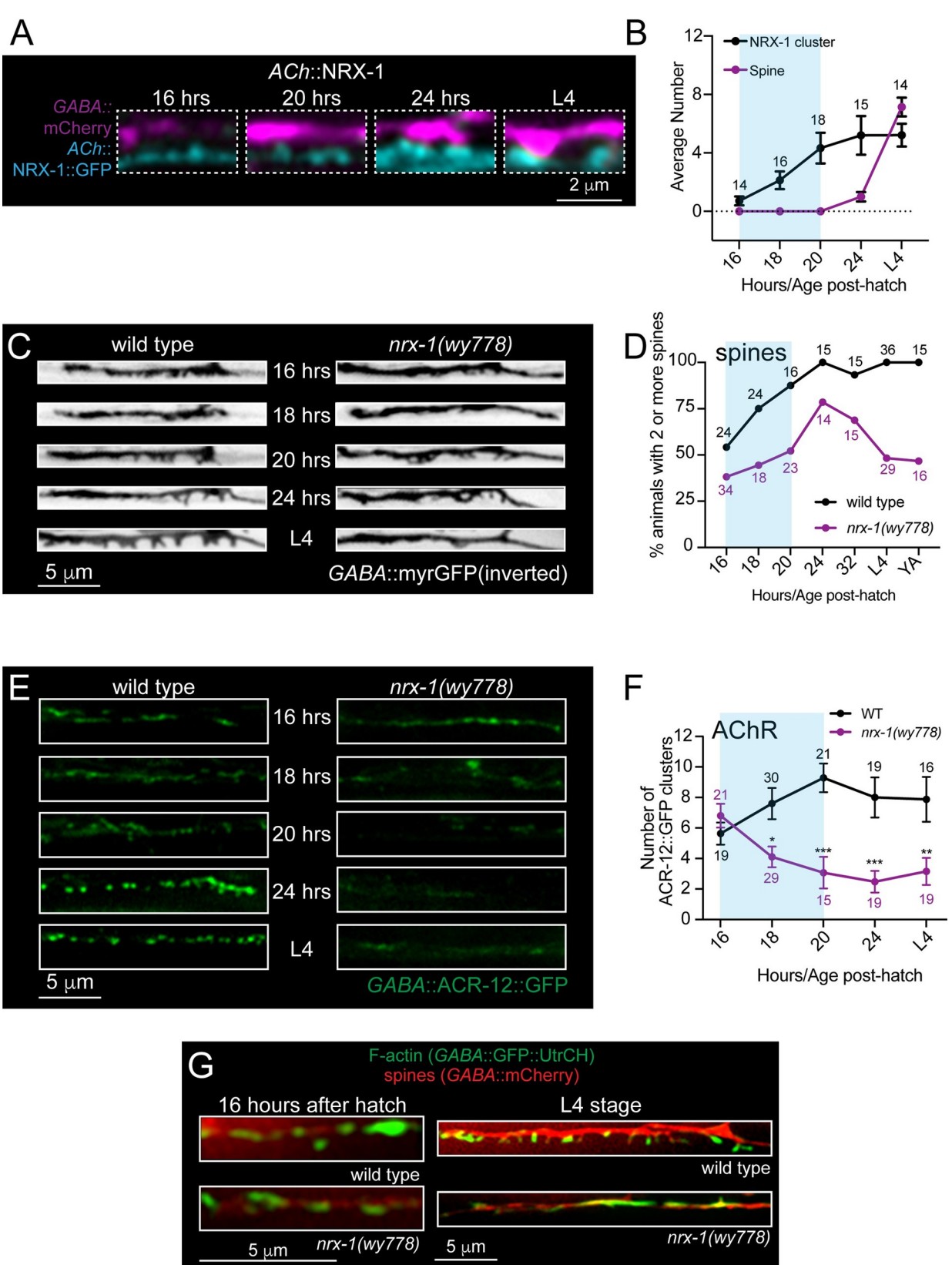

**Fig 3. Dendritic spines and postsynaptic components form initially but subsequently collapse without the synaptic adhesion protein, NRX-1.** (A) Fluorescent confocal images of dendritic spines (P*flp-13*::mCherry, magenta) and cholinergic expression of NRX-1::GFP (P*unc-17β*::NRX-1::GFP, cyan) at 16, 20, and 24 hours after hatch and L4 stage (~42–50 hours post-hatch). Presynaptic clusters of NRX-1::GFP are visible by 16 hours after hatch and increase in size until roughly 24 hours after hatch. (B) Quantification of the average number of spines (purple) and NRX-1 clusters (black) at the time points indicated. Blue shading indicates the approximate period of DD remodeling. Numbers indicate the number of animals quantified at each timepoint. Data points indicate mean ± SEM. (C) Fluorescent images (inverted LUT) of DD dendritic spines (P*flp-13*::myrGFP) in wild type (left) and *nrx-1(wy778)* (right) animals at 16, 18, 20, and 24 hours after hatch and at L4 stage. Immature dendritic spines clusters form initially but then collapse in *nrx-1* mutants. (D) Quantification of the percentage of wild type (black) and *nrx-1(wy778)* (purple) animals with 2 or more DD dendritic spines at the timepoints indicated. YA, young adult. Numbers indicate the number of animals quantified at each timepoint. (E) Fluorescent images of ACR-12/AChR clusters (P*flp-13*::ACR-12::GFP) in DD dendrites of wild type (left) and *nrx-1(wy778)* mutants (right) 16, 20, and 24 hours after hatch and at L4 stage (~42–50 hours post-hatch). Immature AChR clusters initially form but then disperse in *nrx-1* mutants. (F) Quantification of the number of ACR-12::GFP clusters in DD dendrites of wild type (black) and *nrx-1(wy778)* mutants (purple). Two-way ANOVA, Sidak's multiple comparisons, *$p<0.05$, **$p<0.01$, ***$p<0.001$. (G) Fluorescent confocal images of dendritic spines (P*flp-13*::mCherry) and F-actin (P*flp-13*::GFP::UtrCH) in DD dendrites of wild type and *nrx-1(wy778)* animals at 16 hours after hatch and L4 stage.

reduced density relative to wild type (**Figs 3C–3F and S9**). The density of spines in *nrx-1* mutants increased significantly over the next several hours until 24 hours after hatch (**Figs 3C, 3D and S9A**). However, after this time, spine density decreased dramatically such that DD dendrites of *nrx-1* mutants were almost completely devoid of spines by L4 stage [20]. Measurements of spine formation and disassembly in live imaging studies of wild type and *nrx-1* mutants offered further support for this conclusion. Mature wild type spines (L4) were remarkably stable over 1–2 hours of recording, but were more dynamic in the developing circuit (16–20 hrs after hatch) (**S9C–S9I Fig and S3–S5 Videos**). Developing spines in *nrx-1* mutants were also highly dynamic. Notably, *nrx-1* mutant spines exhibited heightened dynamics into later stages of development compared with wild type spines. For example, at 21–24 hours after hatch, almost 90% of wild type spines were stable over the recording period, while less than 50% of *nrx-1* mutant spines remained stable.

Postsynaptic receptor clusters were also present early in the development of *nrx-1* mutants. The number and fluorescence intensity of ACR-12 clusters at 16 hours after hatch in *nrx-1 (wy778)* animals (6.8 ± 0.8 clusters/25 μm) were similar to wild type (5.6 ± 0.7 clusters/25 μm) (**Figs 3E, 3F and S9B**). Within a few hours however, ACR-12 cluster number and intensity decreased significantly in *nrx-1* mutant dendrites, and remained low throughout the remainder of development (**Figs 3E, 3F and S9B**). Remarkably, the organization of dendritic F-actin was similarly affected by *nrx-1* deletion. The distribution of dendritic GFP::UtrCH was comparable across *nrx-1* mutants and wild type in early development (16 hrs after hatch), positioned in discrete clusters along the main dendritic process (**Fig 3G**). By L4 stage, the F-actin marker became almost exclusively associated with wild type dendritic spines, but was diffusely localized along the length of L4 stage *nrx-1* mutant dendrites (**Fig 3G**). Taken together, our results indicate that presynaptic NRX-1 is dispensable for the earliest stages of postsynaptic assembly and spine formation, but is critical for stabilizing dendritic spines, F-actin assemblies and AChR clusters, and promoting their maturation.

## Presynaptic neurexin localizes to cholinergic terminals and stabilizes postsynaptic components in a Kinesin-3/UNC-104 dependent manner

To investigate this model further, we sought to determine how presynaptic neurexin is transported to active zones *in vivo*. Prior work showed that both synaptic vesicles and CLA-1 depend on the Kinesin-3 motor UNC-104 for their delivery to synapses [37,48]. In particular, disruption of *unc-104* causes an accumulation of synaptic vesicles within neuronal somas and a corresponding loss of synaptic vesicles within axons [48] (**S10A and S10D Fig**). *unc-104 (e1265)* carries a D1497N mutation in the PH domain of UNC-104 that impairs cargo binding

and leads to a severe reduction in axonal UNC-104 abundance [49]. We found that NRX-1 endogenously tagged with GFP localized to puncta in neuronal processes of the nerve ring and nerve cords of wild type animals (**S10E Fig**), but was strikingly decreased in the nerve cords of *unc-104(e1265)* mutants (**Figs 4A, 4C** and **S11A–S11D**). A striking deficit in axonal NRX-1:: GFP localization was also evident with specific cholinergic expression of NRX-1::GFP in *unc-104* mutants (**Figs 4B, 4C** and **S11E–S11H**). By contrast, NRX-1::GFP fluorescence was significantly increased (5.4-fold) in *unc-104* mutant cholinergic somas compared to wild type (**S11C, S11D, S11G, S11H Fig**). Consistent with UNC-104 involvement in cholinergic NRX-1 trafficking, we found that UNC-104 and NRX-1 are partially colocalized in cholinergic axons (**S11I and S11J Fig**). Accumulation of NRX-1 in cholinergic cell bodies, coupled with a decrease in axons, indicates a failure of NRX-1 transport in the absence of functional UNC-104. Cholinergic-specific expression of wild type *unc-104* in *unc-104* mutants restored axonal NRX-1::GFP localization, indicating a cell autonomous requirement (**Fig 4B and 4C**). In contrast, mutation of the Kinesin-1 motor *unc-116* did not produce significant accumulation of NRX-1::GFP in cholinergic somas and caused comparatively less severe decreases in axonal NRX-1::GFP, demonstrating preferential involvement of Kinesin-3 for NRX-1 transport (**S12A–S12D Fig**). Live imaging studies offered additional evidence that synaptic vesicle and NRX-1 trafficking may share a common dependence on UNC-104. We found that NRX-1:: GFP trafficking events occurred with similar anterograde and retrograde velocities to SNB-1:: GFP (labeling synaptic vesicles) trafficking events, though less frequently (**S12E–S12I Fig** and **S6, S7** **Videos**). Prior studies have noted that active zone protein trafficking events occur at reduced frequency compared with synaptic vesicle trafficking [9]. Together, our analyses demonstrate that NRX-1 localization to synapses is strongly UNC-104-dependent while UNC-116 may contribute more indirectly, perhaps through effects on other synapse-associated proteins [50,51].

The density of spines and AChR clusters were also severely reduced in L4 stage *unc-104 (e1265)* mutants (**Fig 4D–4F**), similar to *nrx-1* mutants (**Fig 4G**). These effects were rescued by either native or cholinergic expression of wild type *unc-104* in *unc-104(e1265)* mutants, but not by GABA- or muscle-specific *unc-104* expression (**Fig 4D–4F**), demonstrating that UNC-104-mediated transport is critical in presynaptic cholinergic axons for postsynaptic spine development and receptor localization.

We next asked how a failure in delivery of AZ components may contribute to the severe reductions in spines and receptor clusters we observed in *unc-104* mutants. In addition to CLA-1 [37], transport of both ELKS-1/ELKS/CAST and UNC-10/RIM were severely disrupted by mutation of *unc-104*, indicating that Kinesin-3 mediated transport is key for the synaptic delivery and assembly of several active zone constituents in these neurons (**Figs 4H–4J** and **S13**). Remarkably, we found that only mutation of *nrx-1* produced significant decreases in dendritic spines (**Fig 4G**), demonstrating that presynaptic ELKS, UNC-10, and CLA-1 are dispensable for postsynaptic development. Our results demonstrate a specific requirement for NRX-1 in the stabilization and maturation of postsynaptic structures and provide evidence that a failure in synaptic delivery of NRX-1 is a primary causal factor in the postsynaptic structural defects of *unc-104* mutants.

## The stabilization of mature dendritic spines requires ongoing synaptic delivery of NRX-1

Our results suggest UNC-104 mediated transport positions NRX-1 at the presynaptic terminal in the early stages of synapse formation where it acts to stabilize growing postsynaptic structures, including spines and receptor clusters. We next sought to address whether there is a

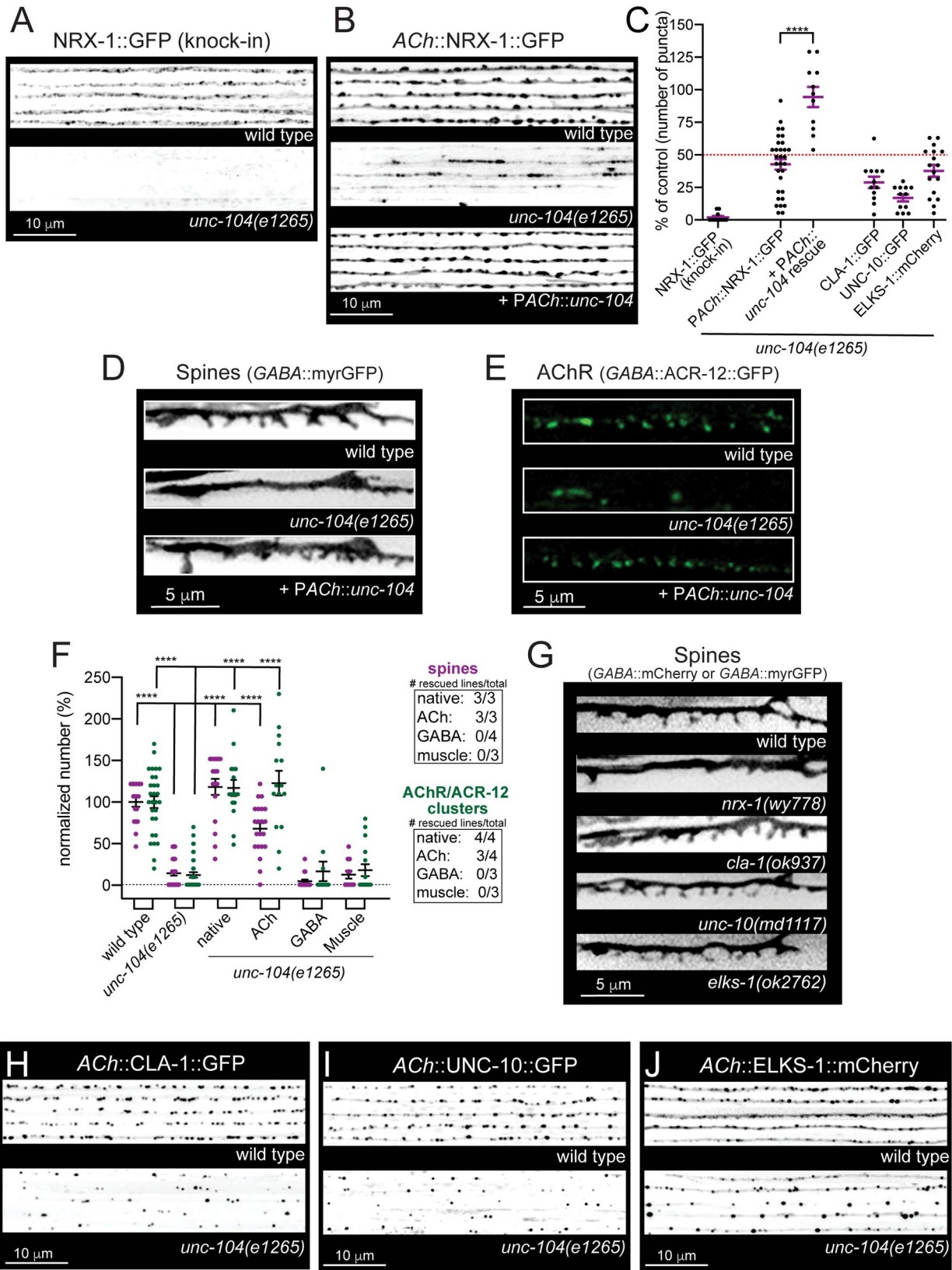

**Fig 4. Presynaptic NRX-1 localizes to cholinergic terminals and stabilizes spines in an UNC-104/KIF1A-dependent manner.** (A) Representative confocal images of NRX-1::GFP (endogenous *nrx-1* knock-in) from dorsal cords (inverted LUT) of wild type and *unc-104(e1265)* mutants. Images on each line are from different young adult animals (5 shown for each genotype). Mutation of *unc-104* impairs axonal NRX-1: GFP localization. (B) Representative confocal images of cholinergic NRX-1::GFP (P*unc-129*::NRX-1::GFP, inverted LUT) from the dorsal nerve cord of wild type, *unc-104(e1265)* and *unc-104(e1265)* mutants rescued by cholinergic expression of wildtype *unc-104* (P*unc-17*β). Images on each line are from different young adult animals (5 shown for each genotype). (C) Scatterplot of the average number of NRX-1::GFP (endogenous or transgenic), CLA-1::GFP, UNC-10::GFP, or ELKS-1::mCherry clusters per 50 μm of dorsal nerve cord in *unc-104(e1265)* mutants or rescue as indicated. Values are normalized to their respective controls. Red dotted line indicates 50% of wild type values. Bars indicate mean ± SEM. Student's t-test, ****$p<0.0001$. n ≥ 11 animals. Raw values with respective controls are shown in S13 Fig. (D) Fluorescent images (inverted LUT) of DD dendritic spines (P*flp-13*::myrGFP) from L4 stage wild type, *unc-104(e1265)*, and *unc-104(e1265)* mutants rescued by cholinergic expression of wild type *unc-104*. (E) Fluorescent images of AChR/ACR-12 clusters in DD dendrites (P*flp-13*::ACR-12::GFP) of L4 stage wild type, *unc-104(e1265)*, and *unc-104(e1265)* mutants rescued by cholinergic expression of wild type *unc-104*. (F) Scatterplot showing quantification of DD dendritic spines per 15 μm (P*flp-13*::myrGFP) (purple) and DD cholinergic receptor clusters (P*flp-13*::ACR-12::GFP) (green) in wild type, *unc-104(e1265)*, and indicated cell-specific rescue lines. Inset, number of rescuing lines/total transgenic lines tested for each rescue construct. Bars, mean ± SEM. One-way ANOVA, Dunnett's multiple comparisons test, ****$p<0.0001$. n ≥ 10 animals. (G) Fluorescent images of DD dendritic spines (P*flp-13*::myrGFP or P*flp-13*::mCherry) from L4 stage wild type, *nrx-1(wy778)*, *cla-1(ok937)*, *unc-10(md1117)*, or *elks-1(ok2762)* mutants. Only mutation of *nrx-1* affects dendritic spines. (H) Representative confocal images of cholinergic CLA-1 (P*unc-17*β::CLA-1::GFP, inverted LUT) from the dorsal nerve cord of wild type or *unc-104(e1265)* mutants. Images on each line are from different young adult animals (5 shown for each genotype). (I) Representative confocal images of cholinergic UNC-10 (P*unc-129*::UNC-10::GFP, inverted LUT) from dorsal nerve cords of wild type and *unc-104(e1265)* mutants. Images on each line are from different young adult animals (5 shown for each genotype). (J) Representative confocal images of cholinergic ELKS-1 (P*unc-129*::ELKS-1::mCherry, inverted LUT) from dorsal nerve cords of wild type and *unc-104(e1265)* mutants. Images on each line are from different young adult animals (5 shown for each genotype).

requirement for presynaptic UNC-104-dependent transport in the stabilization of mature spines. To address this question, we used the previously characterized temperature-sensitive allele, *unc-104(ce782)*. *unc-104(ce782)* animals carry a G105E missense mutation in the motor domain of UNC-104 [52]. When grown at the permissive temperature (13.5˚C), *unc-104 (ce782)* axonal synaptic vesicle abundance and animal motility are modestly reduced compared to wild type. In contrast, synaptic vesicles are completely absent from *unc-104(ce782)* axons following growth at restrictive temperature (20–25˚C), and animal motility is severely compromised within 12 hours of a shift to 23˚C [52]. We raised *unc-104(ce782)* animals to L4 stage at the permissive temperature (13.5˚C). We then shifted L4 stage animals to the restrictive temperature (25˚C) for 16–20 hrs, and quantified NRX-1::GFP localization and spine density immediately following this shift **(Fig 5)**. Axonal NRX-1::GFP clusters were strikingly decreased in *unc-104(ce782)* mutants subjected to the temperature shift compared with control animals subjected to the same shift or *unc-104(ce782)* mutants raised continuously at the permissive temperature **(Fig 5B, 5C)**. *unc-104(ce782)* mutants grown at the permissive temperature had a slightly reduced number of spines overall compared with wild type **(Fig 5E)**. A shift to the restrictive temperature at L3 or L4 stage produced a striking reduction in spine density for *unc-104(ce782)* mutants **(Figs 5D, 5E and S14)**, but not for wild type. Taken together, our findings indicate an ongoing requirement for UNC-104 transport that extends well beyond the period of initial synapse formation and spine outgrowth. Among the UNC-104 cargoes we investigated, only deletion of *nrx-1* produces a significant reduction in spine density at L4 stage. We therefore propose a model where ongoing UNC-104 delivery of presynaptic NRX-1 is critical for postsynaptic maturation and maintenance of mature spines.

## Discussion

### *C. elegans* GABAergic motor neurons are decorated with functional dendritic spines

Dendritic spines are known to act as specialized sites for compartmentalizing neurotransmission and are widely observed across various classes of neurons in both vertebrates and invertebrates [53]. Though suggested by prior EM studies, the presence of functional dendritic spines on *C. elegans* neurons had not been fully appreciated until recently [20,22,23]. Here we

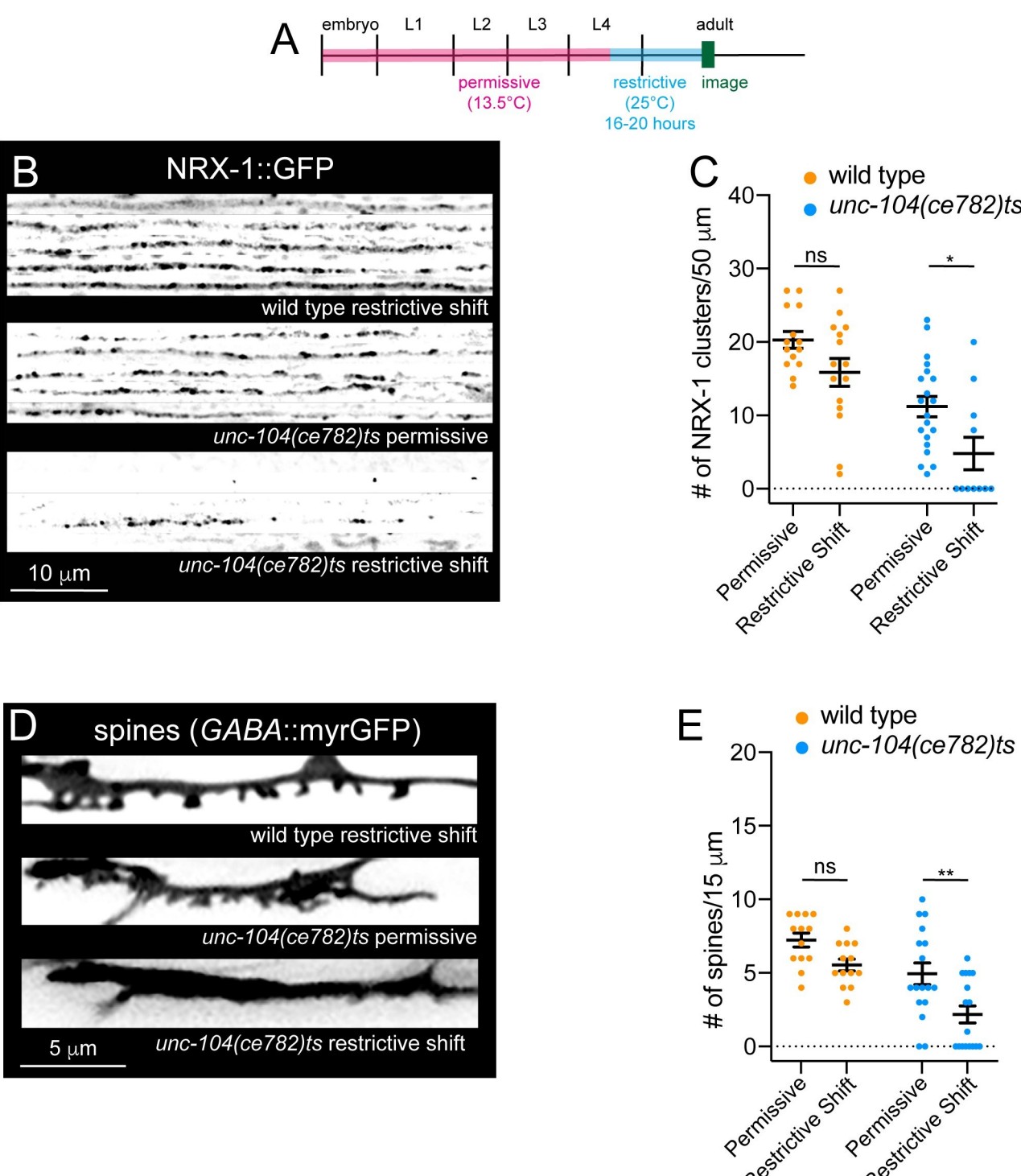

**Fig 5. UNC-104/KIF1A transport of NRX-1 is required in the mature circuit to maintain postsynaptic structure.** (A) Cartoon depiction of experimental timeline. Animals were grown at 13.5°C until L4 stage before shifting animals to the restrictive temperature of 25°C for 16–20 hours and imaging. (B) Line scan fluorescent images of NRX-1::GFP (knock-in to endogenous locus, inverted LUT) from dorsal cords of wild type and *unc-104 (ce782)* mutants grown continuously at the permissive temperature (13.5°C) (middle panel) or shifted to the restrictive temperature (25°C) (bottom panel) at L4 stage for 16–20 hours before imaging (bottom panel). Each line scan indicates an individual animal. (C) Quantification of NRX-1::GFP clusters along a 50 μm region of the dorsal nerve cord in wild type and *unc-104(ce782)ts* animals. Two-way ANOVA, Tukey's multiple comparisons test, ns, not significant, $^*p < 0.05$, $n \geq 11$ animals. Data points indicate mean ± SEM. (D) Fluorescent images of DD spines in wild type and *unc-104 (ce782)ts* animals grown continuously at the permissive temperature (13.5°C) (middle panel) or shifted to the restrictive temperature (25°C) (bottom

panel) at L4 stage for 16–20 hours before imaging. (E) Quantification of the number of spines per 15 μm in wild type and *unc-104(ce782)ts* animals. Two-Way ANOVA, Tukey's multiple comparisons test, ns, not significant, $^{**}p<0.001$. n $\geq$ 13 animals. Data points indicate mean ± SEM.

examined the developmental trajectory of postsynaptic structures (dendritic spines and cholinergic receptor clusters) located on GABAergic motor neurons relative to presynaptic release sites and showed that the stabilization of growing dendritic spines requires presynaptic NRX-1/neurexin.

Neural circuit organization and performance is dictated by the locations of synaptic connections and the identities of the interconnected neurons. Elucidation of the processes governing the construction of synapses and their coordination across presynaptic axons and postsynaptic dendrites are therefore fundamental for understanding circuit assembly and function. However, relatively few studies have monitored in real-time the coordinated assembly of pre- and postsynaptic specializations *in vivo*. Our work tackled this important question, taking advantage of the GABAergic DD spine model and the cellular precision offered by the *C. elegans* motor circuitry. We find that *C. elegans* DD dendritic spines share many of the hallmark features of vertebrate dendritic spines, but also have some key differences. Most notably, mature *C. elegans* spines display limited overt activity-dependent dynamics in wild type animals and appear comparatively stable relative to spines in rodents. This raises the important question of what is the functional role of spines in GABAergic neurons? Prior studies have provided evidence for compartmentalized calcium signals in *C. elegans* neurons [54]. Therefore, one intriguing possibility is that, similar to the situation in mammals, spines serve to compartmentalize calcium and perhaps other biochemical signals in GABAergic neurons. Spines may also be required in order to achieve the unusual dyadic arrangement of synapses in the ventral nerve cord, where presynaptic specializations of cholinergic motor neurons are positioned for transmission onto both GABA motor neuron and muscle postsynaptic partners [55]. In this case, GABAergic spines may have developed to intercept cholinergic release sites onto muscles, as suggested previously [56]. Although not yet characterized by light microscopy, prior electron microscopy studies indicated the presence of spine-like structures in other *C. elegans* neurons such as RMD, RME, SMD, and RIP [21,24,56]. Additional investigation of these structures should help to elucidate their precise functional roles.

We found that F-actin is compartmentalized to discrete regions within the dendritic shafts of DD neurons very early in the development of the circuit (prior to the appearance of spines) and later becomes exclusively localized to dendritic spines. F-actin labeling of spines is apparent as soon as they are detectable, suggesting that F-actin assemblies may participate in the earliest stages of spine development. Prior to remodeling of DD neurons, sites of GABA release are located on the ventral processes of DD neurons. As F-actin based structures are often associated with release sites [57,58], our finding that F-actin is clustered in ventral DD processes prior to remodeling may suggest that F-actin initially associated with release sites persists in the DD neurite through remodeling and is then repurposed for postsynaptic development, perhaps acting as a landmark or actin-based mechanism for stimulating postsynaptic maturation.

Notably, we also observed punctate cholinergic receptor fluorescence in GABAergic dendrites several hours prior to the formation of dendritic spines. These immature receptor clusters colocalize with dendritic F-actin assemblies, suggesting that receptors are trafficked into dendrites and positioned with F-actin to rapidly populate growing spines. Indeed, we observed that receptor clusters are visible in growing spines as soon as spines can be clearly resolved. This is consistent with recent findings from time-lapse 2-photon imaging of hippocampal organotypic slice cultures showing that receptor accumulation in spines occurs concurrently

with spine outgrowth [12]. Similarly, we found that synaptic material accumulates in presynaptic axons prior to the emergence of spines from dendritic processes. In particular, initial accumulations of the presynaptic scaffold CLA-1 and the synaptic vesicle marker synaptobrevin/SNB-1 were visible in the axon with a similar time-course to the appearance of dendritic cholinergic receptor clusters. Presynaptic material became more clearly localized to discrete puncta over the next 4–6 hours, occurring roughly coincident with receptor accumulation at the tips of growing spines. Notably, our findings do not support a strong requirement for presynaptic neurotransmitter release in spine formation.

## Transsynaptic NRX-1/Neurexin signaling stabilizes dendritic F-actin assemblies to promote postsynaptic maturation

Numerous studies across a variety of model systems have examined roles for neurexins in synapse development and function. However, no single consensus view of neurexin function has emerged. Instead, functions for neurexins at specific synapses are thought to be dictated by cellular and molecular context [19,59]. Mammalian genomes encode three Nrxn genes that can give rise to α-, β-, and γ-Nrxn isoforms. These isoforms share common intracellular and transmembrane regions but differ in their extracellular domains. α-Nrxn has six extracellular LNS domains interleaved with three EGF-like repeats. β-Nrxn has only one LNS domain, while γ-Nrxn lacks all identifiable extracellular domains. The multi-faceted roles for neurexins at vertebrate synapses likely emerge as a consequence of the many neurexin isoforms that are generated by alternative splicing, the complexity of their cellular expression, and the potential for these isoforms to selectively interact with distinct postsynaptic partners.

The *C. elegans* genome encodes a single neurexin gene, *nrx-1*, which generates long α and short γ isoforms [60]. As is the case in other systems, *C. elegans* NRX-1 has roles in presynaptic organization, for example in calcium channel clustering at the active zone [47]. Interestingly, the short γ-NRX-1 isoform that lacks identifiable ectodomains performs these roles (worms do not encode β-NRX-1), but even in the absence of both isoforms, a level of presynaptic functionality is maintained, as indicated by the presence of evoked responses in muscle cells following presynaptic motor neuron stimulation [20]. Our prior studies pointed toward the importance of NRX-1 at synapses between cholinergic and GABAergic motor neurons [20]. Surprisingly, our time course studies here revealed that initial spine outgrowth and AChR clustering occur normally in *nrx-1* mutants lacking both isoforms. However, spines and AChRs in *nrx-1* mutants become destabilized within hours and disperse, such that GABAergic dendrites are almost completely devoid of these postsynaptic components by L4 stage. Dendritic F-actin assemblies, that normally show discrete localization to spines, are also disorganized in *nrx-1* mutants, and are diffusely distributed throughout GABAergic dendritic processes in *nrx-1* mutants. We propose a model where presynaptic NRX-1 is required for the stabilization of dendritic spines and other postsynaptic structures in GABAergic dendrites. We suggest that presynaptic NRX-1 directs the organization of dendritic F-actin to stimulate maturation and stabilization of spines and postsynaptic receptor clusters. A similar form of F-actin based reorganization is a key step in presynaptic differentiation [25,57]. Our findings contrast with previous culture studies that implicated neurexin in postsynaptic induction [61].

Our work has parallels with recent studies in *Xenopus* and *Drosophila*. In embryonic *Xenopus* brain, presynaptic β-neurexin stabilizes dendritic filopodia through an adhesive partnership with neuroligin to direct the development of dendritic arbors [62]. Similarly, during fly metamorphosis, neurexin/neuroligin-based adhesion promotes the growth of neurite branches independently of synaptic activity (Constance et al., 2018) [63]. In the present studies, we showed NRX-1 stabilization of dendritic spines occurs independently of synaptic activity.

Previous studies of *nlg-1* reporters [64,65] and more recent transcriptome profiling efforts [66] show that *nlg-1* is not strongly expressed in GABAergic motor neurons. Consistent with these findings, we previously showed deletion of neuroligin/*nlg-1 did not* significantly alter either dendritic spines or receptor clusters in GABAergic neurons at L4 stage [20]. These results suggest NRX-1 interaction with alternate binding partners in GABAergic neurons that to date remain unidentified. Identifying such partners will be an interesting area for future studies.

## UNC-104/Kinesin 3-dependent localization of NRX-1/Neurexin is an essential early step required for synapse maturation and stabilization

Efficient trafficking of synaptic and active zone cargoes to presynaptic terminals is essential for synapse formation and neuronal communication. Long-range axonal transport of synaptic vesicles and active zone proteins is carried out by anterograde kinesin and retrograde dynein motors. However, significant questions remain about which synaptic proteins are trafficked together and their dependence on specific motors. We showed NRX-1 localization to presynaptic terminals is dependent on kinesin-3 UNC-104/KIF1A. The anterograde transport of synaptic and dense core vesicles are also strongly dependent upon UNC-104 [48,67]. In contrast, several studies suggest that other AZ proteins exhibit weaker requirements for UNC-104 in their delivery. For example, ELKS-1 and other AZ proteins localize to *C. elegans* HSN synapses in an UNC-104-independent manner [6]). Similarly, UNC-10/RIM is properly concentrated in nerve ring processes of *unc-104(e1265)* mutants [68]. Here, we found that NRX-1, UNC-10/RIM, ELKS-1/ELKS and CLA-1/piccolo all show a dependence on UNC-104 for anterograde transport to cholinergic synapses. The *unc-104(e1265)* allele used in our study carries a point mutation in the PI(4,5)P2 binding pocket of the PH domain and reduces the ability of UNC-104 to bind cargo [49], perhaps implicating this domain in UNC-104-dependent localization of these proteins.

Prior studies of rodent cultured neurons also suggested a requirement for the Kinesin-3 motor in neurexin transport, indicating that mechanisms for NRX-1 delivery to synapses are conserved [69]. We found that dendritic spines collapse and AChR clusters disperse when UNC-104 delivery of synaptic cargoes is disrupted. Of the putative UNC-104 cargoes we analyzed, only mutation of *nrx-1* produced significant disruption of dendritic spines. We therefore propose that NRX-1 is a key UNC-104 cargo required for spine stabilization. Our live imaging points to similar rates of transport for synaptic vesicles and NRX-1. However, it remains unclear whether NRX-1 is delivered as a component of synaptic vesicles or may be segregated into a distinct vesicular population, such as synaptic vesicle protein transport vesicles (STVs). Our findings that mutations which severely impair synaptic vesicle fusion do not affect spine density point toward the latter possibility.

The prior identification of a temperature-sensitive allele of *unc-104* allowed us to explore the temporal requirements for UNC-104 delivery of NRX-1. Importantly, we found that UNC-104 delivery was not solely required during early stages of synapse maturation but was critical throughout the developmental progression of the circuit toward maturity. These findings argue that delivery of presynaptic NRX-1 is required to maintain postsynaptic structure long after circuit assembly is complete. This raises interesting questions about the relationship between NRX-1 transport and synapse stability, perhaps suggesting that alterations in the rate of axonal NRX-1 transport may directly impact synaptic connectivity in mature animals. More broadly, our studies provide a new view of adhesive mechanisms in circuit connectivity and highlight a novel role for neurexin in the stabilization of mature synapses and dendritic spines.

## Materials and methods

### Strains

All strains are N2 Bristol strain derivatives (wild type) and were maintained at room temperature (20–24°C) on nematode growth media plates (NGM) seeded with *E. coli* strain OP50. Transgenic strains were obtained by microinjection to achieve transformation [70] and identified using co-injection markers. Integrated lines were produced with X-ray irradiation and outcrossed to wild type/N2 Bristol eight times. Only hermaphrodites (L1-L4, young adults) were used in this study. A complete list of all strains used in this study is found in S1 File. Worms used for time course studies were staged by transferring freshly hatched larvae to seeded OP50 plates and transferring to 25°C (time point 0).

### Molecular biology

Plasmids were constructed using the two-slot Gateway Cloning system (Invitrogen) and confirmed by restriction digest and sequencing.

**Utrophin/F-actin reporter.** Sequence coding for mCherry was amplified from pDest-16 (and ligated into AgeI-HF/BspEI-digested PNYL183 (P*flp-13*::GFP::UtrCH, gift from Dong Yan's laboratory) to make pCL87 (P*flp-13*::mCherry::UtrCH).

**TBA-1/tubulin reporter.** Sequence coding for GFP::TBA-1 was amplified from plasmid pYJ128 (gift from Kang Shen laboratory) and ligated into a destination vector to create pDest-173. pDest-173 was recombined with pENTR-5'-*flp-13* to generate pDO61 (P*flp-13*::GFP::TBA-1).

**Golgi apparatus reporter.** pDest-161 (AMAN-2::GFP) was recombined with pENTR-3'-*flp-13* to generate pDO47 (P*flp-13*::AMAN-2::GFP). pDest-161 was created by HindIII-HF digestion of P*gpa-4*::AMAN::GFP (gift from Gert Jansen's laboratory) and ligated with a destination vector.

**Mitochondria reporter.** pre-su9::GFP was isolated from XmaI/SphI-HF digested pDM1389 (gift from A.V. Maricq lab) and ligated into a destination vector to create pDest-91 (pre-su9::GFP). pDest-91 was recombined with pENTR-5'-*flp-13* to generate pAP102 (P*flp-13*::pre-Su9::GFP).

**ER reporter.** Tag-RFP::TRAM-1 was amplified from plasmid pCT27 (gift from A.V. Maricq laboratory) and ligated to a KpnI-HF/NgoMIV digested destination vector to generate pDest-169. pDest-169 was recombined with pENTR-3'-*flp-13* to generate pDO59 (P*flp-13*::RFP::TRAM-1).

**Presynaptic vesicle reporter.** *acr-5* promoter was amplified from plasmid pDM806 and ligated into pENTR-D-TOPO to generate pENTR-40. pENTR-40 was then recombined with pDest-5 to generate pAP264 (P*acr-5*::SNB-1::GFP).

**B-type specific motor neuron reporter.** pDO45 *(Pacr-5::GFP)* was created by recombining pDest-94 (GFP) and pENTR-40 (*acr-5* promoter).

**myrGCamp6f.** myrGCaMP6f was amplified from plasmid pDest-164 and ligated into a destination vector to create pDest-180. pDest-180 was recombined with pENTR3'-*flp-13* to create pDO69 (P*flp-13*::SL2::myrGCamp6f).

**CLA-1 reporter.** 3XnovoGFP::CLA-1e was isolated from SphI-HF/AgeI-HF digested pKP85 (gift from Peri Kurshan) and ligated with a destination vector to create pDest-307 pDest-307 was recombined with pENTR-3'-*unc-17β* to generate pSR58 (P*unc-17β*::3XnovoGFP::CLA-1e cDNA).

**unc-104 rescue constructs.** UNC-104cDNA::mCherry was isolated from SphI-HF/NheI-HF digested pLL48 (gift from Kang Shen lab) and ligated with a destination vector to

make pDest-308. pDest-308 was recombined with pENTR-*unc-17β* to generate pDO126 (P*unc-17β*::UNC-104cDNA::mCherry), pENTR-3'-*unc-47* to generate pDO134 (P*unc-47*:: UNC-104cDNA::mCherry), and pENTR-3'-*myo-3* to generate pDO136 (P*myo-3*::UNC-104cDNA::mCherry).

**P*unc-17β*::NRX-1::GFP.**   pAP112 was generated by recombining pENTR-3'-*unc-17β* with pDest-99 (NRX-1::GFP).

**ELKS-1::mCherry.**   mCherry::ELKS-1cDNA was isolated from SphI-HF/Ascl digested pK043 (gift from Peri Kurshan laboratory) and ligated with a destination vector to create pDest-325. pDest-325 was recombined with pENTR-*unc-129* to generate pDO156 (P*unc*-129:: ELKS-1::mCherry).

**UNC-10::GFP.**   UNC-10::GFP was isolated from BamHI-HF/SbfI digested pRIM3 (gift from Michael Nonet laboratory) and ligated with a destination vector to generate pDest-304. pDest-304 was recombined with pENTR-3'-*unc-129* to generate pDO128 (P*unc-129*::UNC-10::GFP).

**CRISPR/Cas-9 endogenously tagged NRX-1::GFP.**   Strain PHX3578 *nrx-1(syb3578)* was generated in N2 animals by SunyBiotech. Linker sequence and GFP were inserted after exon 27 at the 3' end of C29A12.4a1/*nrx-1*. Sequence flanking the GFP is below.

5': CGGGAAT<u>GG**A**GT**C**</u>GCAAAGAAAAAGGATTTTAAAGAGTGGTACGTAaa
    ggtaccgcgggccc

gggatccaccggtcgccaccatggtg

3': TCCATTTCTTCAATCAAAACTCAATACAATGATGATTAAAAAATTCACTTTTGTCTGCAAA

TTGCCACAACTTCAAAACGGGTAC

PAM sequence is <u>underlined</u>

Synonymous mutations are **bolded**

Linker is in lower case letters

*nrx-1* gene is CAPTIALIZED

## Hydrogel solution

Hydrogel was prepared as an aqueous solution of polyethylene glycol (20% w/v) and the photoinitiator Irgacure (0.5% stock, 0.1% working concentration), and stored at 4˚C in the dark.

## Confocal microscopy

All strains were immobilized with sodium azide (0.3 M) on a 2% or 5% agarose pad. Images were obtained using an Olympus BX51WI spinning disk confocal equipped with a 63x objective. For time course analyses, newly hatched larvae were transferred to a seeded OP50 plate and maintained at 25˚C.

**Long term live imaging of dendritic spines and F-actin.**   Nematodes were immobilized using 2 μL 50 mM muscimol in 10 μL 20% PEG hydrogel solution. Once paralyzed, hydrogel was curated using a handheld UV Transilluminator (312 nm, 3 minutes). Z-stacks were acquired every 5 minutes for at least one hour.

**Live imaging of synaptic vesicle and NRX-1 trafficking.**   Nematodes were immobilized using 50 mM muscimol on a 5% agarose pad. Cholinergic commissures were imaged using Perkin Elmer spinning disk confocal equipped with a 63x objective using 100 ms exposure for 30 seconds.

## Confocal microscopy analysis

All image analysis was conducted using ImageJ software (open source) within defined ROIs using intensity threshold values determined from control experiments. ROIs were located 15–30 μm anterior to DD1, DD2 or DD3 somas.

**Spine analysis.**   Mature spines were quantified as protrusions from the dendrite >0.3 μm in length (measuring from the base to the tip of the protrusion). Spine density was defined as the number of spines/unit length within a selected ROI.

3D rendering was conducted using Imaris/bitplane 3D image analysis software. Morphological categories were determined based on criteria used in [71]. The ImageJ line tool was used to measure the length and width of the spine 3D rendering. Based on these calculations, spines were placed into one of four morphological categories based on the following criteria: Stubby: middle width>total length; Thin: middle width = tip width; Mushroom: tip width>middle width; Branched: >1 spine head.

## Fluorescence intensity analysis

**F-actin/spine and F-actin/tubulin.**   The DD neurite was extracted using the line and line straighten functions. Total fluorescence intensity (utrophin/UtrCH, tubulin/TBA-1, ER/TRAM-1) on the dendritic shaft (line width 4) and below the dendritic shaft (line width 8) were measured. For soma fluorescence intensity, ROI was drawn around the perimeter of the soma.

**Synaptic marker and receptor cluster.**   Background fluorescence was subtracted, and the number and size of synaptic/receptor puncta were measured using the 'analyze particles' function. Confocal montages were assembled using the 'straighten to line' function in a 50 μm region of the dorsal nerve cord.

## Live Imaging

An event is defined as an incidence of spine destabilization/pruning, elongation/formation, or a stable spine during an entire live imaging recording. Z-stacks of DD1 dendrites were acquired every 5 minutes for 60–160 minutes. Numbers indicates number of events for each genotype at each timepoint.

## Calcium Imaging

Animals were grown on plates seeded with OP50 containing 2.7 mM All-Trans Retinal (ATR). Plates were stored at 4°C under dark conditions and used within one week. Imaging was carried out using L4 animals immobilized in hydrogel [72]. Animals were transferred to 7.5 μL of the hydrogel mix placed on a silanized glass slide and covered with a glass slide. Hydrogel was cured using a handheld UV Transilluminator (312 nm, 3 minutes). Post-curing, the covering slide was removed and replaced with a coverslip. Imaging was carried out using a Yokogawa CSU-X1-A1N spinning disk confocal system (Perkin Elmer) equipped with EM-CCD camera (Hamamatsu, C9100-50) and 63X oil immersion objective. Chrimson photoactivation (~30 mW/cm$^2$) was achieved using a TTL-controlled 625 nm light guide coupled LED (Mightex Systems). A 556 nm BrightLine single-edge short-pass dichroic beam splitter was positioned in the light path (Semrock) (**S3 Fig**). Data were acquired at 10 Hz for 15 s using Volocity software and binned at 1×1 during acquisition. Analysis was performed using ImageJ. The DD neurite process in each time series was extracted using the straighten function, background subtracted and photobleaching correction was carried out by fitting an exponential function to the data (CorrectBleach plugin). A smoothing function was applied to the data to enhance signal-to-noise. Individual spine ROIs were identified using the mCherry fluorescence. Post imaging

processing, pre-stimulus baseline fluorescence ($F_0$) was calculated as the average of the data points in the first 4 s of the recording. Data was normalized to prestimulus baseline as $\Delta F/F_0$, where $\Delta F = F-F_0$. Peak $\Delta F/F_0$ was determined by fitting a Gaussian function to the $\Delta F/F_0$ time sequence using Multi peak 2.0 (Igor Pro, WaveMetrics). All collected data were analyzed, including failures (no response to stimulation). Control recordings were carried out in the absence of Retinal.

## Electron microscopy

Staged L4 wild type (P*flp-13*::ACR-12::GFP) hermaphrodites were subjected to high-pressure freeze fixation using a HPM10 high-pressure freezer. Hermaphrodites were slowly freeze substituted in 2% osmium tetroxide (OSO4), 0.1% uranyl acetate, 2% H2O in acetone fixative [73,74]. Samples were embedded in blocks, sectioned at 70 nm-thick serial sections and subsequently collected on copper slot grids. Samples were post-stained in 2.5% uranyl acetate in 70% methanol for 20 minutes and 1:6 lead citrate in $H_2O$ for 2 minutes at room temperature. Images were acquired using the Philips CM10 TEM.

We identified and characterized 6 DD1 spines from a 35 μm region of the ventral head region where the neurite emerges from the cell body of DD1 GABAergic motor neuron.

## Temperature shift experiments

Wild type and *unc-104(ce782)ts* animals eggs were raised at 13.5°C until L3 or L4 stage as indicated, then shifted to 25°C for 16–20 hrs prior to imaging. Animals were age-matched at the time of the temperature shift to account for developmental delays of *unc-104(ce782)ts* mutants, approximately 144 hours to L4 stage for *unc-104(ce782)ts* mutants, and 120 hours for wild type.

## Supporting information

**S1 File. Strain list.**
(XLSX)

**S1 Fig. Cell biology characterization of DD dendrites and spines.** (A) Fluorescent image (top, inverted LUT) and 3D rendering (middle and insets) of DD dendritic spines from an animal expressing P*flp-13*::mCherry, shows diverse spine morphologies. DD dendritic spines share morphological similarities with mammalian dendritic spines: mushroom (5.49 ± 0.66%), branched (5.49 ± 0.66%), stubby (73.63 ± 3.09%), and thin (15.38 ± 1.21%). Arrowheads, spine insets. n = 83 dendritic spines from 11 animals, measurements, percentage ± SD. (B) Fluorescent images of tubulin (P*flp-13*::GFP::TBA-1) and F-actin (P*flp-13*::mCherry::UtrCH) in DD neurons. White arrows indicate dendritic regions where tubulin contacts the base of a dendritic spine. (C) Fluorescent images of DD1 dendritic spines (P*flp-13*::mCherry) and Golgi marker (P*flp-13*::AMAN-2::GFP) in DD neurons. Golgi fluorescence is primarily restricted to the DD1 soma. Dotted white line traces the neuronal cell body. (D) Fluorescent images of DD1 dendritic spines (P*flp-13*::mCherry) and mitochondria (P*flp-13*::pre-Su9::GFP). Mitochondrial fluorescence is visible in the main dendritic processes of DD neurons near spines. (E) Quantification of mitochondria fluorescence intensity in dendritic shaft and dendritic spines. Bars, mean ± SEM. Student's t-test, ****$p<0.0001$. (F) Fluorescent images of DD3 dendritic spines (P*flp-13*::myrGFP) and rough endoplasmic reticulum (ER) (P*flp-13*::RFP::TRAM). TRAM fluorescence is enriched in DD cell bodies but is also more weakly visible in the dendritic processes of DD neurons. (G) Fluorescent images of DD1 dendritic spines (P*flp-13*::myrGFP) and rough endoplasmic reticulum (ER) (P*flp-13*::RFP::TRAM). Fluorescence

signal intensity was increased to better display process.
(TIF)

**S2 Fig. Ultrastructure of DD dendritic spines.** (A) Cartoon depiction of the anterior serial cross-sections used for electron microscopy studies of the ventral nerve cord, focusing on the dendrite of the DD1 neuron. Cross-sections in B are from White and colleagues [21] while sections in C and D are from this work. (B) Serial cross-sections (443-446/597) of the ventral nerve cord from N2U series [21]. Teal indicates DD1 dendrite and dendritic spines dipping into the ventral nerve cord to meet presynaptic cholinergic terminals (magenta). Asterisks indicates mitochondrion within the dendritic shaft, see S1 Fig. (C) Ventral nerve cord electron micrographs. Serial cross-sections (225-228/491) of the ventral nerve cord. Teal indicates DD1 dendrite and dendritic spines (226/491 and 227/491) dipping into the ventral nerve cord to meet presynaptic cholinergic terminals (presumably VA/VB neurons, magenta). Asterisks indicates mitochondria within the dendritic shaft, see S1 Fig. Scale bar, 100 nm. (D) Two representative micrographs from serial sections where the entire extent of the spine is visible are shown. Asterisk indicates mitochondrion within the dendritic shaft. Scale bar, 100 nm.
(TIF)

**S3 Fig. Cartoon illustration of calcium imaging recording platform.** Imaging was performed using a Yokogawa CSU-X1-A1N spinning disk confocal system (Perkin Elmer) equipped with EM-CCD camera (Hamamatsu, C9100-50) and 63X oil immersion objective. Chrimson photoactivation ($\sim$30 mW/cm$^2$) was achieved using a TTL-controlled 625 nm light guide coupled LED (Mightex Systems), permitting illumination of the entire immobilized animal, while simultaneously recording GCaMP6f fluorescence (excitation 488 nm, emission 525 nm). A 556 nm edge BrightLine single-edge short-pass dichroic beam splitter was positioned in the light path (Semrock) to prevent 625 nm light from reaching the camera.
(TIF)

**S4 Fig. The anterior to posterior development of post-embryonic born ventral cholinergic motor neurons and GABAergic dendritic spines.** (A) Fluorescent images of B-type cholinergic neurons (DB/VB) (P*acr-5*::GFP) (green) and DD GABAergic neurons (P*flp-13*::mCherry) (red) at 16 and 24 hours after hatch. VB cholinergic neurons are born post-embryonically in an anterior to posterior order (blue). Note that at 16 hours after hatch P*acr-5*::GFP fluorescence indicating VB1 and VB2 cell bodies is visible. By 24 hours after hatch P*acr-5*::GFP fluorescence indicating VB3 is visible. White dotted circles outline the neuronal cell bodies. (B) Fluorescent images of the anterior ventral nerve cord in animals co-expressing the synaptic vesicle marker P*acr-5*::SNB-1::GFP in B-type cholinergic neurons (DB/VB) (green) with P*flp-13*::mCherry labeling DD GABAergic neurons (red) at 12, 16, and 24 hours after hatch. At 12 hours, little SNB-1::GFP fluorescence is visible in the ventral nerve cord. Embryonic born, dorsally directed B-type (DB) cholinergic motor neurons predominantly make synaptic contacts in the dorsal nerve cord. Ventrally-directed, B-type (VB) cholinergic motor neurons are born post-embryonically and have not yet completed their maturation at this time. SNB-1::GFP fluorescence in the ventral nerve cord is faintly visible by 16 hrs after hatch and becomes more prominent by 24 hrs after hatch, coincident with maturation of VB motor neurons. White dotted circles indicate outlines of neuronal cell bodies. Yellow arrows indicated presynaptic vesicle clusters, SNB-1. (C) Cartoon representation of the developmental timing of post-embryonic born ventral cholinergic motor neurons, spine outgrowth, and cholinergic synaptic vesicle localization. (D-E) Fluorescent images (inverted LUT) of DD1 and DD2 dendritic spines 24 hours after hatch (D) and at L4 ($\sim$42–50 hours after hatch) stage (E). Animals express P*flp-13*::mCherry to label dendritic spines. Red dashed rectangle indicates inset anterior to

DD1 soma. Blue dashed rectangle indicates inset anterior to DD2 soma. Note that DD1 spines form prior to DD2 spines. (F) Quantification of the length of growing dendritic spines at 24 and 32 hours after hatch and at L4 (~42–50 hours after hatch) stage. Data points indicate mean ± SEM.
(TIF)

**S5 Fig. Development of pre- and postsynaptic components.** Fluorescent images of GABAergic dendrites at 12, 16, 20, 24, and 32 hours after hatch and at L4 stage (~42–50 hours posthatch). Animals express a GABAergic dendrite marker (P*flp-13*::mCherry) with either (A) presynaptic active zone marker (P*unc-17β*::GFPnovo2::CLA-1e), (B) synaptic vesicle marker (P*acr-5*::SNB-1::GFP) or postsynaptic receptors (C) (P*unc-47*::ACR-12::GFP) or (D) F-actin (P*flp-13*::GFP::UtrCH) (P*unc-17β*::GFPnovo2::CLA-1e). Images are pseudo colored to indicate spines (magenta) and synaptic components (CLA-1, vesicle clusters, F-actin, or AChRs) (cyan).
(TIF)

**S6 Fig. LEV-10 localization and actin dynamics during development.** (A) Fluorescent confocal images showing dendritic spines (magenta) and the postsynaptic CUB transmembrane domain LEV-10 (cyan). Animals express P*flp-13*::mCherry with DD neuron specific LEV-10::GFP11$_{x7}$ using a strategy for cell-specific labeling of endogenous LEV-10 (NATF) [39]. (B) Quantification of the number of DD dendritic spines and LEV-10 clusters at 16, 20, and 24 hours after hatch, and L4 stage (~42–50 hours post-hatch). (C) (C, E, G) Line scans displaying relative fluorescence intensity of F-actin (P*flp-13*::GFP::UtrCH) at 16 hours after hatch (C), 24 hours after hatch (E), and L4 (~42–50 hours after hatch) animals (G). Each color indicates a line scan of fluorescence intensity for the same DD dendrite ROI acquired at 5-minute intervals. Note the variable distribution of fluorescence intensities across line scans from images acquired near 16 hours after hatch compared to later time points, indicating increased F-actin dynamics during early development. (D, F, H) Confocal images (inverted LUT) showing P*flp-13*::GFP::UtrCH fluorescence (labeling F-actin) in the DD dendrite of 16 (D), or 24 (F) hours after hatch, or at L4 stage (~42–50 hours after hatch) (H). For each, sequences of fluorescent images separated by 30 minutes are shown.
(TIF)

**S7 Fig. Spines and AChR localize independently of presynaptic activity.** (A, B) Fluorescent confocal images (inverted LUT) of DD1 dendritic spines (P*flp-13*::mCherry (A) or P*flp-13*::myrGFP (B)) in L4 stage wild type and selected mutant strains where synaptic activity is affected. (A) Fluorescent confocal images of AChR clusters (P*flp-13*::ACR-12::GFP) in the DD dendrite of L4 stage wild type and selected mutant strains where synaptic activity is affected. (B) Fluorescent confocal images (inverted LUT) of DD1 dendritic spines (P*flp-13*::myrGFP) in L4 stage wild type and *nrx-1(wy1155)* null mutants. (C) Quantification of the number of DD1 dendritic spines in L4 wild type and *nrx-1(wy1155)* null mutants. Student's t-test, ****$p < 0.0001$. Bars, mean ± SEM.
(TIF)

**S8 Fig. Dendritic calcium responses are severely reduced in *nrx-1* mutants.** Scatter plot showing peak $\Delta F/F_o$ responses measured from GABAergic DD motor neuron dendrites during a 5s period of cholinergic photostimulation in wild type, *acr-16(ok789); acr-12(ok367)*, *nrx-1(wy778)*, *nrx-1(nu485)*, and *nrx-1(ok1649)* mutants. All genotypes co-express P*flp-13*::myrGCaMP6f::SL2::mCherry for measurement of dendritic calcium responses with P*acr-2*::Chrimson for cholinergic neuron depolarization. Dendritic calcium responses were not significantly affected by deletion of the homomeric nAChR subunit *acr-16*, but were significantly

reduced by mutation of either *acr-12* or *nrx-1*. Bars indicate mean ± SEM. One-way ANOVA, Dunnett's multiple comparisons, ****$p<0.0001$. Wild type control is the same as Fig 1H. n ≥ 10 animals.
(TIF)

**S9 Fig. Spine and AChR localization and spine dynamics in developing wild type and *nrx-1 (wy778)* mutants.** (A) Quantification of the number of DD spines/15 μm in wild type (black) and *nrx-1(wy778)* (purple) animals 16, 20, 24, 32 hours after hatch and at L4 (~42–50 hours after hatch) and young adult (YA) (~52–56 hours after hatch) stages. Two-way ANOVA, Sidak's multiple comparisons test, * $p<0.05$, **$p<0.01$, ****$p<0.0001$. Data points indicate mean ± SEM. Numbers indicate animals quantified for each timepoint. (B) Quantification of ACR-12::GFP fluorescence intensity from DD dendrites of wild type (black) and *nrx-1(wy778)* (purple) animals at 16, 20, and 24 hours after hatch and L4 stage (~42–50 hours post-hatch). Two-way ANOVA, Sidak's multiple comparisons test, **$p<0.01$, ***$p<0.001$. Data points indicate mean ± SEM. (C) Fluorescent image (inverted LUT) of DD dendritic spines at timepoint zero in a wild type animal 16–20 hours after hatch. Dashed red box indicates region shown in S9D Fig. (D) Fluorescent images (inverted LUT) of an individual wild type DD dendritic spine (P*flp-13*::mCherry) at 0, 45, and 70 minute timepoints during live imaging of area indicated by red box in S9C Fig. Red dashed line indicates largest extent of spine outgrowth. (E) Fluorescent image (inverted LUT) of *nrx-1(wy778)* mutant DD1 dendritic spines (P*flp-13*::mCherry) at timepoint zero acquired 16–20 hours after hatch. Dashed red and blue boxes indicate regions shown in S9F Fig. (F) Fluorescent images (inverted LUT) of individual *nrx-1(wy778)* mutant DD dendritic spines (P*flp-13*::mCherry) at (top) 90. 95, and 100 minute timepoints during live imaging of area indicated by red box in S9E Fig and (bottom) at 555, 70, 75, 90 minute timepoints during live imaging of area indicated by blue box in S9E Fig. Red dashed line indicates largest extent of spine outgrowth. Red arrows indicate spine dynamics. (G) Fluorescent image (inverted LUT) of wild type DD dendritic spines (P*flp-13*::mCherry) at timepoint zero at L4 stage. Dashed red box indicates region shown in S9H Fig. (H) Fluorescent images (inverted LUT) of an individual wild type DD dendritic spine (P*flp-13*::mCherry) at 0, 30, and 65 minute timepoints during live imaging of the area indicated by red box in S9G Fig. (I) Quantification of the percentage of spine events in wild type and *nrx-1(wy778)* animals at 16–20 hours after hatch, 21–24 hours after hatch, and at L4 (~42–50 hours after hatch) compared to the total number of events. An event is defined as an incidence of spine destabilization/pruning, elongation/formation, or a stable spine during an entire live imaging recording. Z-stacks of DD1 dendrites were acquired every 5 minutes for 60–160 minutes. Numbers indicates number of events for each genotype at each timepoint.
(TIF)

**S10 Fig. Mutation of *unc-104* impairs synaptic vesicle delivery to cholinergic axons in the dorsal nerve cord and NRX-1 endogenously tagged with GFP localizes to neuronal processes.** (A) Fluorescent images of cholinergic synaptic vesicles (P*acr-2*::mCherry::RAB-3) in the dorsal nerve cord of young adult wild type and *unc-104(e1265)* mutants. Images on each line are from different animals (4 are shown for each genotype). (B) Quantification of cholinergic synaptic vesicles (P*acr-2*::mCherry::RAB-3) per 50 μm of the dorsal nerve cords of wild type and *unc-104(e1265)* mutants. Student's t-test, ****$p<0.0001$. Bars, mean ± SEM. (C) Fluorescent images of the soma in wild type and *unc-104(e1265)* mutants expressing cholinergic vesicle reporter (P*acr-2*::mCherry::RAB-3). Dotted white lines outline the cell body. (D) Quantification of soma cholinergic synaptic vesicle fluorescence intensity (AU) (P*acr-2*::mCherry:: RAB-3) of wild type and *unc-104(e1265)* mutants. Student's t-test, ****$p<0.0001$. Bars, mean ± SEM. (E) NRX-1::GFP localizes within neuronal processes of the nerve ring and shows

punctate localization in processes of the ventral and dorsal nerve cords of L4 stage worms.
(TIF)

**S11 Fig. NRX-1 localization to cholinergic axons is decreased in *unc-104* mutants and colocalizes with UNC-104 in cholinergic axon terminals.** (A) Quantification of NRX-1::GFP (endogenous knock-in) clusters in a 50 μm region of the dorsal nerve cord in wild type and *unc-104(e1265)* animals. Bars, mean ± SEM. Student's t-test, ****$p<0.0001$. n $\geq$ 11 animals. (B) Quantification of NRX-1::GFP (endogenous knock-in) axon fluorescence intensity in a 50 μm region of the dorsal nerve cord. Bars, mean ± SEM. Student's t-test, ****$p<0.0001$. n $\geq$ 11 animals. (C) Fluorescent images of NRX-1::GFP (endogenous knock-in) in somas of wild type and *unc-104(e1265)* mutants. Dotted white lines outline the neuronal cell body. (D) Quantification of NRX-1::GFP soma fluorescence intensity. Bars, mean ± SEM. Student's test, ****$p<0.001$. (E) Quantification of the number of NRX-1::GFP (P*unc129*::NRX-1::GFP) clusters in a 50 μm region of the dorsal nerve cord in wild type, *unc-104(e1265)*, and *unc-104 (e1265)* mutants rescued with cholinergic expression of wild type *unc-104*. Bars, mean ± SEM. One-way ANOVA, Dunnett's multiple comparisons test, ****$p<0.0001$. n $\geq$ 11 animals. (F) Quantification of NRX-1::GFP fluorescence intensity in cholinergic axons (P*unc129*::NRX-1::GFP) of a 50 μm region of the dorsal nerve cord. Bars, mean ± SEM. One-way ANOVA, Dunnett's multiple comparisons test, ****$p<0.0001$. n $\geq$ 11 animals. (G) Fluorescent images of NRX-1::GFP in cholinergic somas of wild type, *unc-104(e1265)* and *unc-104(e1265)* mutants rescued with cholinergic expression of wild type *unc-104*. Dotted white lines outline the neuronal cell body. (H) Quantification of NRX-1::GFP fluorescence intensity in cholinergic somas. Bars, mean ± SEM. One-way ANOVA, Dunnett's multiple comparisons test, ****$p<0.001$. (I) Line scans depicting relative fluorescent intensity of NRX-1::GFP (green) and UNC-104::mCherry (red) for a 50 μm region of the dorsal nerve cord. Gray dotted rectangles indicate corresponding puncta in S11J Fig. (J) Fluorescent images of the dorsal nerve cord in an adult animal expressing NRX-1::GFP (P*unc-129*::NRX-1::GFP) and UNC-104::mCherry (P*unc-129*:: UNC-104::mCherry). Colocalization is indicated by white carets.
(TIF)

**S12 Fig. NRX-1 localizes independently of UNC-116/KIF5A/C and has similar anterograde and retrograde velocity as SNB-1.** (A) Fluorescent images (inverted LUT) of NRX-1::GFP (P*unc-129*::NRX-1::GFP) from the dorsal nerve cord of wild type and *unc-116(e2310)* mutants. Images on each line are from different animals (5 are shown for each genotype). (B) Quantification of the number of NRX-1 clusters (P*unc-129*::NRX-1::GFP) in a 50 μm region of the dorsal nerve cord of wild type and *unc-116(e2310)* animals. Bars, mean ± SEM. Student's t-test, ***$p<0.001$. n $\geq$ 14 animals. (C) Fluorescent images of NRX-1 (P*unc-129*::NRX-1::GFP) from cholinergic neuron somas of wild type and *unc-116(e2310)* mutants. Dotted white lines outline cell bodies. (D) Quantification of fluorescence intensity of NRX-1 (P*unc-129*::NRX-1::GFP) from cholinergic somas of wild type and *unc-116(e2310)* mutants. Bars, mean ± SEM. Student's t-test, ns, not significant. (E, F) Kymographs of synaptic vesicle (P*unc-129*::SNB-1::GFP) **(E)** and NRX-1 (P*unc-129*::NRX-1::GFP) **(F)** trafficking events recorded from cholinergic neuron commissures. (G) Quantification of the anterograde velocity (μm/second) of SNB-1::GFP (blue) and NRX-1::GFP (orange) along the axonal commissure. ns, not significant. n $\geq$ 10 animals for all panels. (H) Quantification of the retrograde velocity (μm/second) of SNB-1::GFP (blue) and NRX-1::GFP (orange) along the axonal commissure. ns, not significant. (I) Quantification of the total number of SNB-1::GFP (blue) and NRX-1::GFP (orange) trafficking events binned into retrograde and anterograde directions. Note that NRX-1 trafficking events occur significantly less frequently than SNB-1 events.
(TIF)

**S13 Fig. Active zone protein localization in *unc-104(e1265)* mutants.** (A) Scatterplot of CLA-1::GFP puncta number in a 50 μm region of the dorsal nerve cord in wild type and *unc-104(e1265)* animals. Student's t-test, ****$p < 0.0001$. Bars, mean ± SEM. These data correspond to quantification in Fig 4C. (B) Scatterplot of CLA-1::GFP fluorescence intensity in a 50 μm region of the dorsal nerve cord in wild type and *unc-104(e1265)* animals. Student's t-test, ****$p < 0.0001$. Bars, mean ± SEM. (C) Scatterplot of UNC-10::GFP puncta number in a 50 μm region of the dorsal nerve cord in wild type and *unc-104(e1265)* animals. Student's t-test, ****$p < 0.0001$. Bars, mean ± SEM. These data correspond to quantification in Fig 4C. (D) Scatterplot of UNC-10::GFP fluorescence intensity in a 50 μm region of the dorsal nerve cord in wild type and *unc-104(e1265)* mutants. Student's t-test, ****$p < 0.0001$. Bars, mean ± SEM. (E) Scatterplot of ELKS-1::mCherry puncta number per 50 μm of the dorsal nerve cord in wild type and *unc-104(e1265)* animals. Student's t-test, ****$p < 0.0001$. Bars, mean ± SEM. These data correspond to quantification in Fig 4C. (F) Scatterplot of ELKS-1::mCherry fluorescence intensity in a 50 μm region of the dorsal nerve cords of wild type and *unc-104(e1265)* mutants. Student's t-test, ***$p < 0.001$. Bars, mean ± SEM.
(TIF)

**S14 Fig. Shift to restrictive temperature during L3 in wild type and *unc-104(ce782)ts* animals.** (A) Cartoon depiction of experimental timeline. Animals were grown at 13.5°C until L3 stage (~approx. 120 hours in *unc-104(ce782)* mutants, 96 hours in wild type animals) before shifting animals to their restrictive temperature of 25°C for 16–20 hours and imaging. (B) Fluorescent images of DD spines (P*flp-13*::myrGFP) from wild type (top) or *unc-104(ce782)ts* animals grown continuously at the permissive temperature (13.5°C) (middle panel) or shifted to the restrictive temperature (25°C) for 16–20 hours before imaging. (C) Quantification of DD spines per 15 μm from wild type and *unc-104(ce782)ts* mutants. Two-way ANOVA, Tukey's multiple comparisons test, ****$p < 0.0001$, $n \geq 12$ animals. Data points indicate mean ± SEM.
(TIF)

**S1 Video. Confocal live imaging video of F-actin dynamics (labeled by P*flp-13*::GFP:: UtrCH) in the DD1 dendrite of wild type animals at 16 hours after hatch. For display, movies are shown at 3 fps. Images were acquired every 5 minutes.**
(AVI)

**S2 Video. Confocal live imaging video of F-actin dynamics (labeled by P*flp-13*::GFP:: UtrCH) in the DD1 dendrite of wild type animals at L4 stage. For display, movies are shown at 3 fps. Images were acquired every 5 minutes.**
(AVI)

**S3 Video. Wild type DD dendritic spines (P*flp-13*::myrGFP) 16 hours after hatch. A z-stack was acquired every 5 minutes for at least 1 hr of imaging duration.** For display, movies are shown at 3 fps
(AVI)

**S4 Video. *nrx-1(wy778)* mutant DD dendritic spines (P*flp-13*::myrGFP) 16 hours after hatch.** A z-stack was acquired every 5 minutes for at least 1 hr of imaging duration. For display, movies are shown at 3 fps
(AVI)

**S5 Video. Wild type DD dendritic spines (P*flp-13*::myrGFP) at L4 stage. A z-stack was acquired every 5 minutes for at least 1 hr of imaging duration.** For display, movies are

shown at 3 fps
(AVI)

**S6 Video. NRX-1::GFP trafficking in a cholinergic commissure of a young adult animal (P*unc-129*::NRX-1::GFP).** Videos were recorded at 100 ms exposure for 30 seconds. For display, video is shown at 25 fps.
(AVI)

**S7 Video. SNB-1::GFP trafficking in a cholinergic commissure of a young adult animal (P*unc-129*::SNB-1::GFP).** Videos were recorded at 100 ms exposure for 30 seconds. For display, video is shown at 25 fps.
(AVI)

## Acknowledgments

Nematode strains were provided by the *Caenorhabditis* Genetics Center which is funded by the NIH National Center for Research Resources. Thank you to Dori Schafer for access to 3D rendering Imaris software. Thank you to Dirk Albrecht's laboratory for advice with hydrogel immobilization. Thank you to the laboratories of: Dong Yan, Kang Shen, Andres Maricq, Michael Nonet, Peri Kurshan, and Gert Jansen for providing reagents. Thank you to Peri Kurshan for sharing unpublished results. Thank you to William Joyce and Michael Gorczyca for technical assistance. Thank you to Steve Cook for help with analysis of electron micrographs and John White for use of N2U electron micrographs. Lastly, we thank members of the Francis laboratory for manuscript comments.

## Author Contributions

**Conceptualization:** Devyn Oliver, David H. Hall, Michael M. Francis.

**Data curation:** Devyn Oliver, Shankar Ramachandran, Alison Philbrook, Christopher M. Lambert, Ken C. Q. Nguyen, David H. Hall.

**Formal analysis:** Devyn Oliver, Shankar Ramachandran, Michael M. Francis.

**Funding acquisition:** Michael M. Francis.

**Investigation:** Devyn Oliver, Shankar Ramachandran, Michael M. Francis.

**Methodology:** Devyn Oliver, Shankar Ramachandran, Michael M. Francis.

**Project administration:** Devyn Oliver, Michael M. Francis.

**Resources:** Ken C. Q. Nguyen, David H. Hall, Michael M. Francis.

**Supervision:** Michael M. Francis.

**Validation:** Devyn Oliver.

**Visualization:** Devyn Oliver, Shankar Ramachandran.

**Writing – original draft:** Devyn Oliver, Michael M. Francis.

**Writing – review & editing:** Devyn Oliver, Shankar Ramachandran, Michael M. Francis.

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
