## [Decision Letter · Decision Letter 0]

21 Sep 2021

Dear Dr Francis,

Thank you very much for submitting your Research Article entitled 'Kinesin-3 mediated axonal delivery of presynaptic neurexin stabilizes dendritic spines and postsynaptic components' to PLOS Genetics.

The manuscript was fully evaluated at the editorial level and by independent peer reviewers. The reviewers appreciated the attention to an important problem, but raised some substantial concerns about the current manuscript. Based on the reviews, we will not be able to accept this version of the manuscript, but we would be willing to review a much-revised version. We cannot, of course, promise publication at that time.

If you decide to revise the manuscript for further consideration at PLOS Genetics, please aim to resubmit within the next 60 days, unless it will take extra time to address the concerns of the reviewers, in which case we would appreciate an expected resubmission date by email to plosgenetics@plos.org.

[LINK]

We are sorry that we cannot be more positive about your manuscript at this stage. Please do not hesitate to contact us if you have any concerns or questions.

Yours sincerely,

Fengwei Yu

Associate Editor

PLOS Genetics

Gregory P. Copenhaver

Editor-in-Chief

PLOS Genetics

Reviewer's Responses to Questions

**Comments to the Authors:**

Reviewer #1: This manuscript examines the developmental timeline and molecular mechanisms that control the formation and stabilization of Spine Like Portrusions (SLPs) in C.elegans. Previous work from the Francis lab identified SPLs and provided their initial characterization. The current manuscript follows up on that work. The authors describe the timeline of spine formation, looking at cytoskeletal components and specific organelles. They then go on to focus on nrx-1 (which they have previously shown plays a role in SLP formation. They provide data that Neurexin’s main role is in synapse stabilization, rather than early formation. They also describe Neurexin as an UNC-104/kinesin-3 cargo and suggest that it is actually the main UNC-104 cargo that is important (on the presynaptic side) for SLP maintenance.

Overall, I strongly support publication. The manuscript provides a nice addition to our knowledge of SLP development in C.elegans. While the results of many of the experiments seem more confirmatory than novel, the scope and depth of the work certainly warrant publication in my opinion. In addition, the manuscript is clearly written and overall attention to details is high.

There are several issues which should be addressed:

1. The tubulin marker looks quite diffuse and possibly not integrated into MTs. This is likely because of the C-ter tag. TBA-1 should be tagged on the N terminus. My personal view is that a partial analysis of microtubules (i.e. not studying their dynamics, polarity and organization), even with a better marker, does not contribute significantly to the paper and I would suggest removing it. Otherwise, to argue that the SLPs are similar to spines in terms of the microtubule cytoskeleton the authors would need to image microtubule invasions into spines (see here for example: https://pubmed.ncbi.nlm.nih.gov/27658622/).

2. The marker used to claim presence rough ER in SLPs seems highly overexpressed since it does not show ER pattern in the cell body. Similar to point (1), my view is that since the presence of ribosomes in SLPs was already shown by EM, this supplementary figure detracts rather adds to the paper. If the authors wish to make the claim about rough ER being in SLPs they should show more convincing data and explain why their data contrasts with other work that failed to detect rough ER in worm neurites (i.e Rolls&Rapoport).

3. Can they show a more zoomed-out version of data in Figure 2A and B?

4. Developmental timeline (wildtype versus neurexin): Is it possible to add a timepoint between 24hrs and L4 as well as one timepoint after L4? That would substantiate the claim of a stabilization function for Neurexin.

5. I did not understand the claim that “Receptor clustering followed a similar trend to spine formation” since in Figure 3E ACR-12::GFP seems affected already at 18hrs. Can they explain?

6. The analysis of how activity mutants affect spines (Table 1) is very nice. However, there is some discrepancy (for example unc-13 vs unc-18, where one is significant and the other isn’t although presumably both regulate the same process). Can they add a column that note whether the alleles used nulls or weak/strong hypomorphs?

7. The claims about UNC-104/KIF1A transporting neurexin should be modified to state that Neurexin localizes in an unc-104 dependent manner. To claim transport by unc-104 the authors should show more direct evidence such as co-transport with a known unc-104 cargo.

Reviewer #2: This study by Oliver and colleagues explores the time course of postsynaptic spines at the GABAergic motor neurons in the nematode C. elegans. The findings provide significant insight into an important topic: how spine formation and maturation are determined. By analyzing the presynaptic organization and postsynaptic spine development, the authors presented trajectory of spine formation and maturation. The paper also indicates that cell adhesion molecule nrx-1/neurexin is a key component during spine maturation but is not required for spine formation. Moreover, unc-104/kinesin motor protein also plays important roles by transporting nrx-1 to proper sites. Overall, the study is methodical, experimentally sound, and the manuscript is well-written and logical. The manuscript presents some interesting findings but will however be strengthened considerably by addressing several key points as detailed below.

1. Is the spine structure GABAergic synapses specific in worm? The authors may need to give some background somewhere.

2. Page9. Ca2+ responses in spines were reduced by 80% in unc-17 mutants. What are the 20% remaining? Is this evoked by other neurotransmitters from cholinergic neurons? This question was asked because cholinergic synaptic transmission is completely abolished in the NMJs of the unc-17 mutants.

3. Since spines on the GABAergic motor neurons have different shapes (Figure 1B), how is the spine length measured in Table 1?

4. Page 13. The authors should be careful when saying that the postsynaptic spine formation is not highly related to the presynaptic activity. The mutants that block presynaptic SV exocytosis, such as unc-13 and unc-18, were used. Based on the results from NMJs, a substantial spontaneous release can still occur in unc-13 (e51) and unc-18 mutants. The low frequency spontaneous events may be sufficient to trigger spine formation. The author may need to consider using the unc-13 (s69) mutant which is almost null and no spontaneous release occurs (Hu et al, 2013, eLife). Unc-31 mutant is not a good example as the spontaneous release is unaltered (Gracheva et al., 2007, JNS). The observation in unc-17 mutants is somewhat surprising. Is it possible that other neurotransmitters are also involved? The spine number and length are both significantly reduced in acr-2 gf mutants, but the authors didn’t give an description for the results. In Zhou’s paper (2017, Cell Reports), spontaneous release occurs normally in this mutant. Overall, it will be interesting to know how spine formation is affected in mutants that completely block transmission, or on the contrary, that have increased presynaptic activity.

5. In Figure3A, the number of the NRX-1::GFP puncta is almost the same with the spine number, while the spine number is significantly lower than the presynaptic puncta of CLA::GFP or SNB-1::GFP. The author may need to explain the differences.

6. Is the nrx-1 wy778 a null mutant? If worm also expresses different isoforms of nrx-1, are they all deleted in wy778 allele?

7. Although the authors presented clear evidence that nrx-1 is involved in spine maturation, they didn’t mention the important binding partner of nrx-1, nlg-1/neuroligin. Is nlg-1 localized in the postsynaptic GABAergic spines? It will be interesting to know whether the observed phenotypes in nrx-1 mutants are also nlg-1 dependent.

8. How does unc-104/kinesin transport nrx-1 to presynaptic sites? Do they bind to each other? Do both the long and short isoforms of nrx-1 have the transmembrane domain? Are both isoforms required for spine maturation?

Reviewer #3: PGENETICS-D-21-01188 “Kinesin-3 mediated axonal delivery of presynaptic neurexin stabilizes dendritic spines and postsynaptic components”

Dendritic spines, the small protrusions from the dendrite membrane, are the contact site with neighboring axons where synaptic input is received. Recent advances in microscopy have uncovered that also in C. elegans, spine like protrusions of GABAergic motor neurons exhibit some of the hallmarks of mammalian dendritic spines. In this study, Oliver et al focus on the developmental aspects of the formation of post synaptic specialization sites of GABAergic motor neurons. The authors nicely define the order and timing of the formation of the pre and post synaptic sites, and demonstrate that NRX-1 is required for stabilization, but not initial formation of the dendritic spines. The kinesin UNC-104 transports NRX-1 and its activity is required for synapse maturation and stabilization. The last experiments shown in figure 5 of temperature shifts of the unc-104 mutants nicely teases out the temporal aspects of the requirement of UNC-104 delivery of NRX-1 for postsynaptic maturation and maintenance of mature spines. Overall this is an interesting manuscript with important and thorough findings, that are significant for our understanding of the functional properties of neural circuits. My specific comments can be found below.

My main point goes toward emphasizing the novelty of this paper beyond the previous two recently published studies. Philbrook et al 2018 and Cuentas-Condori et al 2019 both describe the functional aspects of the GABAergic dendritic spines in C. eleagns. Cuentas-Condori defines these structures, the cytoskeletal properties and the calcium transients triggered by presynaptic activity. Philbrook at, besides describing these structures, also demonstrates that NRX-1, located at presynaptic sites, specifically directs postsynaptic development. The authors do go beyond these two manuscripts, but in my opinion often fail to distinguish previous already published findings from new unpublished data. To give one example, Figure 1 is highly redundant with previous studies and I almost do not see anything that wasn’t already described. Why do the authors characterize spine morphology in Figure 1B? it was already characterized in Cuentas-Condori et al and the authors don’t use it again later on in the manuscript (they could, for example in table 1 of receptor analysis). This point could be addressed by some rewriting to highlight novel points and shorten where redundancy occurs.

Minor points:

1. In Figure 1 panel I (calcium imaging), please explain why the Gaussian fitting was used.

2. In Figure 4, please mention the developmental stage of the imaged worms (I assume its L4).

3. In Figure 4, the title of the figure could be rephrased: the evidence presented in this figure support the requirement of nrx-1 to synapse formation or stabilization but not necessarily stabilization.

4. In Figure S4.5, what the authors call ‘modest decrease in axonal NRX::GFP’ in unc-116 mutants, is actually a reduction in a third of the signal. I suggest that the authors at least raise possible explanation as to why the residual activity of UNC-116 does not appear in unc-104 mutants.

5. In Figure 5, it would add extra value and novelty to the Figure if the authors prolonged the experiment and added one additional shift back to the permissive temperature after L4, to address whether de novo formation of NRX-1 synapses is possible during adulthood.

6. The supplemental figure list is extremely long. The authors can easily combine many of the supplementary figures into denser versions, for a better presentation of the figures and a smoother reading.

**Have all data underlying the figures and results presented in the manuscript been provided?**

Reviewer #1: None

Reviewer #2: Yes

Reviewer #3: Yes

PLOS authors have the option to publish the peer review history of their article (what does this mean?). If published, this will include your full peer review and any attached files.

Reviewer #1: No

Reviewer #2: **Yes: **Zhitao Hu

Reviewer #3: No

---

## [Decision Letter · Decision Letter 1]

3 Jan 2022

Dear Dr Francis,

We are pleased to inform you that your manuscript entitled "Kinesin-3 mediated axonal delivery of presynaptic neurexin stabilizes dendritic spines and postsynaptic components" has been editorially accepted for publication in PLOS Genetics. Congratulations!

Yours sincerely,

Fengwei Yu

Associate Editor

PLOS Genetics

Gregory P. Copenhaver

Editor-in-Chief

PLOS Genetics

Comments from the reviewers (if applicable):

Reviewer's Responses to Questions

**Comments to the Authors:**

Reviewer #1: My concerns have been addressed. Thanks!

Reviewer #2: It is usually a good way to include the new added results/discussion/changes in the rebuttal when addressing specific questions. This will help the reviewer to find the right place quickly.

Reviewer #3: The authors have adequately addressed all my concerns to the best of their ability and changed/added the requested modifications to my content. I therefore recommend to accept the manuscript in its current form.

**Have all data underlying the figures and results presented in the manuscript been provided?**

Reviewer #1: None

Reviewer #2: Yes

Reviewer #3: Yes

PLOS authors have the option to publish the peer review history of their article (what does this mean?). If published, this will include your full peer review and any attached files.

Reviewer #1: No

Reviewer #2: **Yes: **Zhitao Hu

Reviewer #3: No

**Data Deposition**

http://datadryad.org/submit?journalID=pgenetics&manu=PGENETICS-D-21-01188R1

**Press Queries**

---

## [Editor Report · Acceptance letter]

17 Jan 2022

PGENETICS-D-21-01188R1 

Kinesin-3 mediated axonal delivery of presynaptic neurexin stabilizes dendritic spines and postsynaptic components 

Dear Dr Francis, 

We are pleased to inform you that your manuscript entitled "Kinesin-3 mediated axonal delivery of presynaptic neurexin stabilizes dendritic spines and postsynaptic components" has been formally accepted for publication in PLOS Genetics! Your manuscript is now with our production department and you will be notified of the publication date in due course.

With kind regards,

Orsolya Voros

PLOS Genetics

On behalf of:
